# Blood–brain barrier genetic disruption leads to protective barrier formation at the Glia Limitans

**Pierre Mora**[1☯], **Pierre-Louis Hollier**[1☯], **Sarah Guimbal**[1☯], **Alice Abelanet**[1], **Aïssata Diop**[1], **Lauriane Cornuault**[1], **Thierry Couffinhal**[1], **Sam Horng**[2], **Alain-Pierre Gadeau**[1], **Marie-Ange Renault**[1], **Candice Chapouly**[1] *

**1** Univ. Bordeaux, INSERM, Biology of Cardiovascular Diseases, Pessac, France, **2** Department of Neurology and Neuroscience, Icahn School of Medicine at Mount Sinai, New York City, New York, United States of America

☯ These authors contributed equally to this work.
* candice.chapouly@inserm.fr

**Data Availability Statement:** All relevant data are within the paper and its Supporting Information files. All S1 Data and S2 Data files are available from the Figshare public repository database. The

## Abstract

Inflammation of the central nervous system (CNS) induces endothelial blood–brain barrier (BBB) opening as well as the formation of a tight junction barrier between reactive astrocytes at the Glia Limitans. We hypothesized that the CNS parenchyma may acquire protection from the reactive astrocytic Glia Limitans not only during neuroinflammation but also when BBB integrity is compromised in the resting state. Previous studies found that astrocyte-derived Sonic hedgehog (SHH) stabilizes the BBB during CNS inflammatory disease, while endothelial-derived desert hedgehog (DHH) is expressed at the BBB under resting conditions. Here, we investigated the effects of endothelial Dhh on the integrity of the BBB and Glia Limitans. We first characterized DHH expression within endothelial cells at the BBB, then demonstrated that DHH is down-regulated during experimental autoimmune encephalomyelitis (EAE). Using a mouse model in which endothelial Dhh is inducibly deleted, we found that endothelial Dhh both opens the BBB via the modulation of forkhead box O1 (FoxO1) transcriptional activity and induces a tight junctional barrier at the Glia Limitans. We confirmed the relevance of this glial barrier system in human multiple sclerosis active lesions. These results provide evidence for the novel concept of "chronic neuroinflammatory tolerance" in which BBB opening in the resting state is sufficient to stimulate a protective barrier at the Glia Limitans that limits the severity of subsequent neuroinflammatory disease. In summary, genetic disruption of the BBB generates endothelial signals that drive the formation under resting conditions of a secondary barrier at the Glia Limitans with protective effects against subsequent CNS inflammation. The concept of a reciprocally regulated CNS double barrier system has implications for treatment strategies in both the acute and chronic phases of multiple sclerosis pathophysiology.

S1 Data DOI link is https://doi.org/10.6084/m9.
figshare.12625034.v6 The S2 Data DOI link is
https://doi.org/10.6084/m9.figshare.12625085.v7.

**Funding:** This study was supported by grants from
the European Council (Marie Skłodowska-Curie
Actions, Individual fellowship 2019 (MSCA-IF-
2019)) (grant number GA794726) and the
Fondation ARSEP (Fondation pour la Recherche
sur la Sclérose En Plaques (https://www.arsep.org/
) (grant number ARSEP 2019/R19083GG). This
study was also co-funded by the "Institut National
de la Santé et de la Recherche Médicale" (project
U103420G, grant number U1034SE20GA). Grant
number GA794726 was received by CC, grant
number ARSEP 2019/R19083GG was received by
CC and grant number U1034SE20GA was received
by CC. The funders had no role in study design,
data collection and analysis, decision to publish, or
preparation of the manuscript. CC received a salary
for 2 years (2019 and 2020) from the European
Council as part of the grant GA794726.

**Competing interests:** The authors have declared
that no competing interests exist.

**Abbreviations:** Aldh1l1, *aldehyde dehydrogenase 1
family*, *member l1*; ALB, albumin; AQP4, aquaporin
4; ABM, astrocyte basal medium; BBB, blood–brain
barrier; BSA, bovine serum albumin; CD45, cluster
of differentiation 45; CDH5, cadherin5; CLDN4,
Claudin4; Cldn5, *claudin5*; CNS, central nervous
system; CTNNβ1, catenin β1; DHH, desert
hedgehog; DMEM, Dulbecco's Modified Eagle
Medium; EAE, experimental autoimmune
encephalomyelitis; EBM-2, endothelial basal
medium-2; FBS, fetal bovine serum; FGB,
fibrinogen; FITC, fluorescein isothiocyanate; FoxO1,
forkhead box O1; GFAP, glial fibrillary acidic
protein; HBMEC, human brain microvascular
endothelial cell; HH, Hedgehog; IBA1, ionized
calcium binding adaptor molecule 1; IgG,
immunoglobulin G; Icam1, *intercellular adhesion
molecule 1*; IHH, Indian hedgehog; IL-1β,
interleukin-1β; IMPC, International Mouse
Phenotyping Consortium; LAM, laminin; LPS,
lipopolysaccharide; MEC, microvascular endothelial
cell; MMP, matrix metalloproteinase; MOG$_{35-55}$,
myelin oligodendrocyte glycoprotein-35-55; NG2,
neural/glia antigen 2; NHA, normal human
astrocytes; PDGFB, platelet-derived growth factor
subunit b; PDGFRβ, platelet-derived growth factor
receptor beta; PECAM1, platelet/endothelial cell
adhesion molecule 1; PTCH1, Patched-1; PTX,
pertussis toxin; PVS, perivascular space; qRT-PCR,
quantitative reverse transcription polymerase chain
reaction; RIPA, radioimmunoprecipitation assay;
SHH, Sonic hedgehog; SMA, smooth muscle actin;
SMO, Smoothened; Vcam1, *vascular cell adhesion*

## Introduction

In a healthy individual, the central nervous system (CNS) parenchyma is protected from the peripheral circulation by the blood–brain barrier (BBB), which tightly regulates the entry and exit of soluble factors and immune cells [1]. Importantly, during multiple sclerosis, the abnormal permeability of the BBB allows penetration into the CNS parenchyma of inflammatory cells and soluble factors such as autoantibodies, cytokines, and toxic plasma proteins, which drive lesion formation and acute disease exacerbation [2,3]. Therefore, identifying key mechanisms that promote BBB tightness is currently considered to be a main strategy for controlling leukocyte and humoral entry, preventing acute relapse and disability progression in multiple sclerosis.

Previous studies have identified the Hedgehog (HH) pathway as a regulator of BBB integrity in multiple sclerosis, HIV, and stroke [4–7]. Desert hedgehog (DHH) is expressed constitutively at the BBB in adults [8] and belongs, together with Sonic hedgehog (SHH) and Indian hedgehog (IHH), to the HH family of morphogens, identified nearly 4 decades ago in Drosophila as crucial regulators of cell fate determination during embryogenesis [9]. The interaction of HH proteins with their specific receptor Patched-1 (PTCH1) derepresses the transmembrane protein Smoothened (SMO), which activates downstream pathways including the canonical HH pathway leading to the activation of Gli family zinc finger (Gli) transcription factors, and the so-called noncanonical HH pathways, which are independent of SMO and/or Gli [10].

Interestingly, a wealth of literature published during the last decades has enabled a change in the vision of BBB structure and integrity, which has expanded to include contributions from both barrier properties of the vascular endothelial cells and the astrocytic end feet of the neurovascular unit. Within the neurovascular unit, substantial intercellular communication network involves the vascular endothelial cells and astrocytic end feet, as well as the pericytes and basement membranes within the perivascular space (PVS) [11–13]. How these signals regulate the passage of soluble factors and cells into and out of the CNS is not completely understood and is of considerable translational interest to the field of neuroimmunology. Regulatory mechanisms at the BBB include solute transporters and receptor-mediated transcytosis, and immune cells are actively prevented from crossing the BBB by low levels of immune receptors that normally permit immune trafficking. Once soluble factors and immune cells penetrate the BBB, they circulate within the PVS, a region between the basal basement membrane of the endothelial cell wall and the parenchymal basement membrane abutting the astrocyte end feet [14,15]. While it is now well established that BBB breakdown leads to soluble factor and inflammatory cell infiltration into the PVS during neuropathology, the role of the Glia Limitans is more complex. Indeed, astrocytes, described as reactive, may demonstrate opposing roles in both recruiting and restricting neuroinflammatory infiltration depending on the context [16]. Specific reactive astrocyte behaviors are likely determined by signaling events that vary with the nature and severity of CNS injury or disease. Specifically, in multiple sclerosis as well as Alzheimer's and Parkinson diseases, it has been shown that reactive astrocytes, on one hand, produce pro-inflammatory and pro-permeability factors and on the other hand, neuroprotective factors [17–19].

Astrocyte barrier properties are not as well characterized as those of the BBB. However, several groups have highlighted barrier properties at the Glia Limitans [20–22]. Notably, endfoot–endfoot clefts, similar to those observed between endothelial cells at the BBB, have been described at the Glia Limitans and shown to be responsible for the sieving effect observed between the distribution of small and large Dextrans [23]. Moreover, under neuroinflammatory conditions, immune cell trafficking across the Glia Limitans is necessary for clinical

molecule 1; VEGFA, vascular endothelial growth factor A; Vim, *vimentin*; WMA, World Medical Association; ZO1, zonula occludens 1.

experimental autoimmune encephalomyelitis (EAE) [21]; indeed, matrix metalloproteinase (MMP)-2 and MMP-9 proteolytically cleave dystroglycan, which anchors astrocyte end feet to the Glia Limitans basement membrane via binding to extracellular matrix molecules, allowing infiltrating leukocytes to penetrate the parenchyma [24]. Additionally, it has been shown that the scavenger receptor CXCR7 is up-regulated on the inflamed BBB endothelium, facilitating mobilization of T cells from the PVS into the CNS parenchyma [25]. Altogether, these data demonstrate that both the endothelial BBB and its basement membrane, along with the Glia Limitans and the parenchymal basement membrane, are required for immune cell trafficking across the neurovascular unit. Strikingly, our recent work has given considerable attention to a new property of reactive astrocytes: the expression of tight junction proteins (notably Claudin4 (CLDN4)) under inflammatory conditions [26]. This result provides yet another argument in favor of astrocytic barrier properties.

The first objective of our study was to decipher the role of the morphogen DHH in maintaining BBB tightness. The second objective was to demonstrate that a double barrier system comprising both the BBB and Glia Limitans is implemented in the CNS and regulated by a crosstalk going from endothelial cell to astrocytes using endothelial Dhh knockdown as a model of permeable BBB.

Here, we first demonstrate that endothelial DHH expression is down-regulated during neuroinflammation and is necessary to maintain BBB tightness. We then show that BBB opening, induced by Dhh knockdown, drives astrocyte CLDN4 expression, conferring barrier properties to the Glia Limitans, which results in the PVS entrapment of plasma proteins and inflammatory cells, both under physiological conditions and during pathology. Together, these data identify the neurovascular unit as a double barrier system whose function is controlled by the crosstalk between endothelial cells and astrocytes.

In conclusion, this work strengthens the concept of CNS double barrier system, unveiling how signals at the endothelium drive astrocyte barrier properties to protect the parenchyma during neuropathology. Consequently, taking into account both components of the neurovascular unit is of translational interest and could open the way for new therapeutic strategies notably to limit progressive multiple sclerosis pathology.

## Results

### DHH, but not SHH or IHH, is expressed by CNS microvascular endothelial cells and down-regulated during chronic neuroinflammation

First, we showed that DHH is expressed at the BBB in vitro using mouse CNS MECs* (Fig 1A). CNS MEC purity was assessed using platelet/endothelial cell adhesion molecule 1 (PECAM1) and zonula occludens 1 (ZO1) as positive endothelial markers and smooth muscle actin (SMA), neural/glia antigen 2 (NG2), cluster of differentiation 45 (CD45), ionized calcium binding adaptor molecule 1 (IBA1), and glial fibrillary acidic protein (GFAP) as markers of contamination by smooth muscle cells, pericytes, leukocytes, microglia, and astrocytes, respectively (Fig 1A and S1A Fig). DHH expression at the BBB was verified in vivo using human cortical sections from healthy donors (Fig 1B) and brain sections from C57BL/6 mice (S1B Fig). SHH and IHH are not expressed in the healthy BBB, and DHH is known to be stored intracellularly as well as being secreted [7]. Therefore, we here infer the CNS endothelial cells as the source of DHH within the neurovascular unit. Next, we demonstrated that *Dhh* is severely down-regulated at the BBB under inflammatory conditions both in vitro using human brain microvascular endothelial cells (HBMECs) treated with interleukin-1β (IL-1β) (one of the main pro-inflammatory cytokine implicated in multiple sclerosis pathophysiology) (Fig 1C and S1 Data) and in vivo (Fig 1F and S1 Data) using a preclinical model of multiple sclerosis

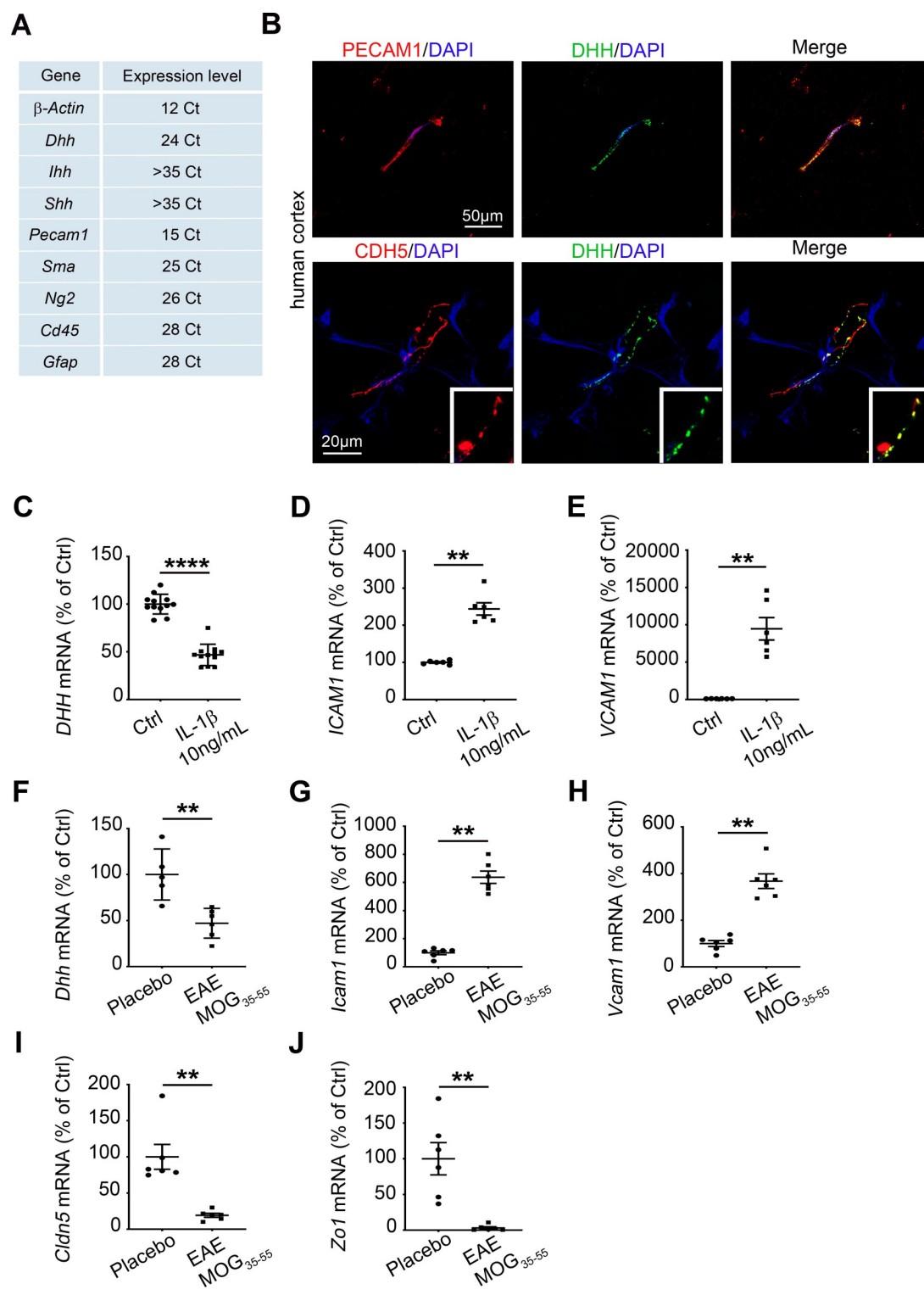

**Fig 1. DHH is expressed by CNS MECs and down-regulated during chronic neuroinflammation. (A)** Primary BMECs were isolated from 12-week-old C57Bl/6 mice, and *Dhh*, *Ihh*, *Shh*, *Pecam1*, *Sma*, *Ng2*, *Cd45*, *and Gfap* expressions were quantified by qRT-PCR (cycle threshold mean values). *β-actin* is used as a reference. **(B)** Human cortical sections from healthy donors were obtain from the NeuroCEB biobank and immunostained with anti-CDH5 (in red), anti-PECAM1 (in red), and anti-DHH (in green) antibodies. Nuclei were stained with DAPI (in blue). **(C–E)** HBMECs were cultured until confluency and starved for 24 h. HBMECs were then treated with PBS (control condition) or IL-1β 10 ng/mL for 24 h and **(C)** *DHH*, **(D)** *ICAM1*, and **(E)**

*VCAM1* expression were quantified by qRT-PCR. **(F–H)** Twelve-week-old C57Bl/6 females (6 animals per group) were induced with MOG$_{35-55}$ EAE versus placebo. At day 13 post induction, mice were humanely killed, and spinal cord microvascular endothelial cells were isolated. **(F)** *Dhh*, **(G)** *Icam1*, **(H)** *Vcam1*, **(I)** *Cldn5*, and **(J)** *Zo1* expression were measured via qRT-PCR in both groups (MOG$_{35-55}$ versus placebo). $^{**}P \leq 0.01$, $^{****}P \leq 0.0001$ Mann–Whitney U test. The underlying data for Fig 1 can be found in S1 Data (https://doi.org/10.6084/m9.figshare.12625034.v6). *It is important to note that CNS endothelial cells are from a pooled source including both brain and spinal cord tissues. Therefore, the resulting cell cultures/lysates may be heterogeneous in their use of DHH. This remark applies to Figs 1–4. BMECs, brain microvascular endothelial cells; CD45, cluster of differentiation 45; CDH5, cadherin5; Cldn5, claudin5; CNS, central nervous system; Ctrl, control; DHH desert hedgehog; EAE, experimental autoimmune encephalomyelitis; GFAP, glial fibrillary acidic protein; HBMECs, human brain microvascular endothelial cells; IHH, Indian hedgehog; cam1, intercellular adhesion molecule 1; IL-1β, interleukin 1 beta; MEC, microvascular endothelial cell; MOG$_{35-55}$, myelin oligodendrocyte glycoprotein-35-55; NS, non-significant; NG2, neural/glia antigen 2; qRT-PCR, quantitative reverse transcription polymerase chain reaction; PECAM1, platelet/endothelial cell adhesion molecule 1; SMA, smooth muscle actin; SHH, Sonic hedgehog; Vcam1, vascular cell adhesion molecule 1; ZO1, zonula occludens 1.

(MOG$_{35-55}$) to induce chronic neuroinflammation in C57BL/6 mice. For this experiment, isolated spinal cord microvessels underwent a digestion step followed by a CD45$^+$ T cell depletion step to discard inflammatory cell contamination induced by EAE (S1C and S1D Fig). *Dhh* down-regulation at the BBB is associated with the up-regulation of endothelial activation markers *intercellular adhesion molecule 1 (Icam1)* (Fig 1D–1G and S1 Data) and *vascular cell adhesion molecule 1 (Vcam1)* (Fig 1E–1H and S1 Data) and with down-regulation of mRNA markers of tight junctions (*claudin5* (*Cldn5*) and *Zo1*) (Fig 1I, 1J and S1 Data).

Together, these data identify DHH as the only HH expressed in adults at the endothelial BBB. Moreover, they highlight the fact that DHH expression is down-regulated at the BBB during neuroinflammatory pathology.

## Endothelial-specific Dhh inactivation induces down-regulation of adherens junction CDH5 and tight junction CLDN5 ex vivo

To test the importance of endothelial DHH expression at the BBB, we conditionally disrupted DHH expression in endothelial cells and examined the consequences on BBB integrity. To do so, we used CNS MEC cultures isolated from *Cdh5-Cre$^{ERT2}$*, *Dhh$^{Flox/Flox}$* mice (Dhh$^{ECKO}$ mice), and *Dhh$^{Flox/Flox}$* control littermates, 2 weeks after inducing knockdown by intraperitoneal injection of tamoxifen. There is no difference between Dhh$^{ECKO}$ and control mouse cell culture viability (S1E, S1F Fig and S2 Data).

We first verified the efficiency of the knockout by measuring *Dhh* expression in primary CNS MEC cultures obtained from Dhh$^{ECKO}$ and littermate controls and showed that *Dhh* expression is strongly down-regulated in the knockout mice (Fig 2A and S1 Data). Moreover, CDH5, CLDN5, and ZO1 junctions are disorganized in Dhh$^{ECKO}$ endothelial cells: In controls, CDH5, CLDN5, and ZO1 display a well-defined pattern of sharp contours at endothelial cell–cell contacts. In contrast, in Dhh$^{ECKO}$ cultures, a broader, more irregular pattern is detected at endothelial cell–cell contacts (Fig 2D–2F). This result is consistent with the previously documented phenotype using small interfering RNA (siRNA) for DHH [8]. CDH5 and CLDN5 but not ZO1 are down-regulated in Dhh$^{ECKO}$ CNS MECs compared to controls (Fig 2B, 2C and 2G–2I and S1 Data).

We concluded that endothelial DHH expression is necessary to maintain endothelial adherens and tight junction mRNA and protein expression level at the BBB and to maintain a well-defined CDH5, ZO1, and CLDN5 pattern of sharp contours at endothelial cell–cell contacts.

## DHH induces CLDN5 up regulation through the inhibition of FOXO1 transcriptional activity in vitro

We previously identified DHH as a factor facilitating the interaction between CDH5 and catenin β1 (CTNNβ1) in endothelial cells [8], and others have shown that CDH5 interacts with

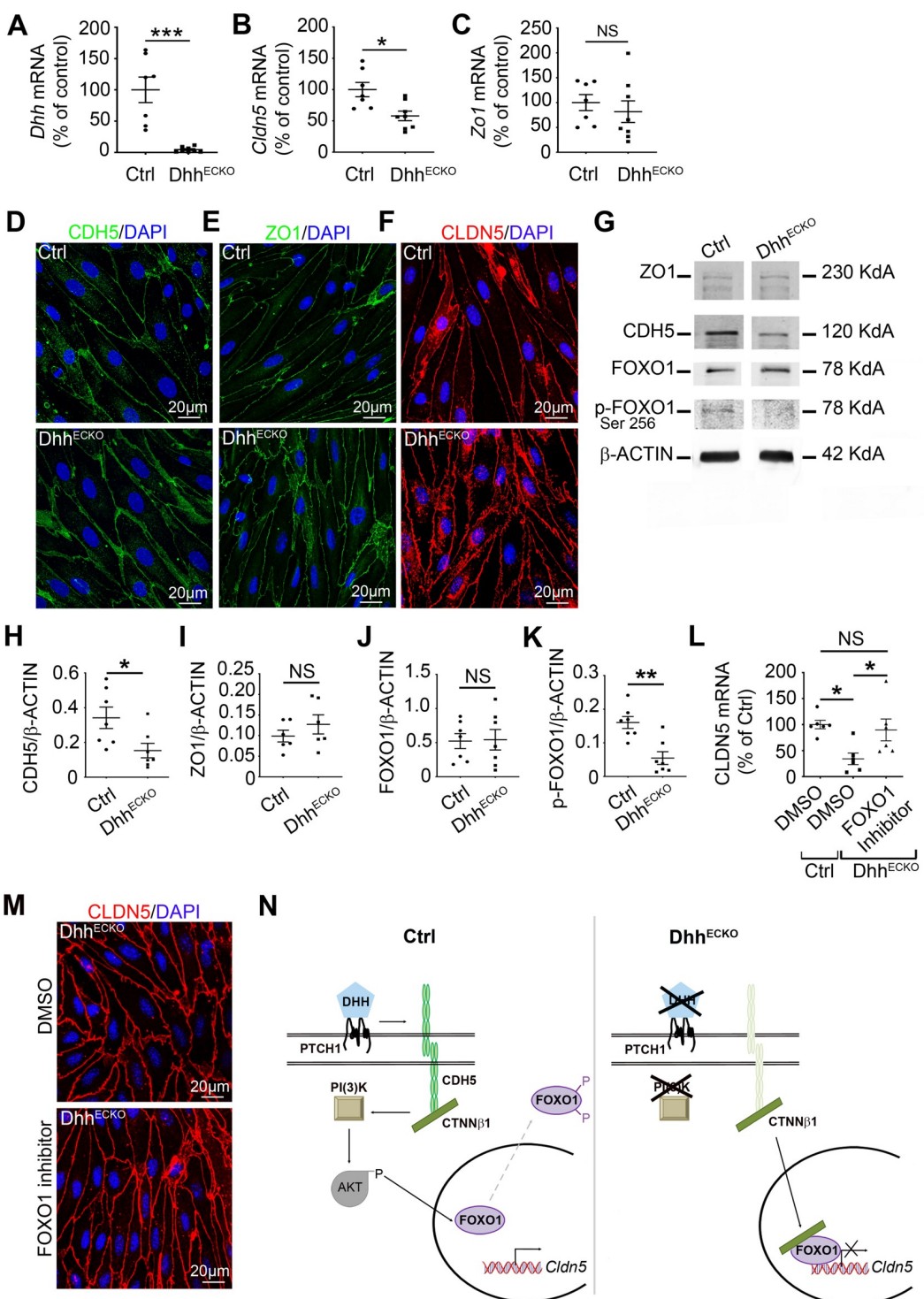

**Fig 2. Endothelial-specific *Dhh* inactivation induces adherens and tight junction down-regulation through stimulation of FOXO1 transcriptional activity.** (**A–C**) CNS MECs were isolated from Dhh[ECKO] and control mice, and (**A**) *Dhh*, (**B**) *Cldn5*, and (**C**) *Zo1* expressions were quantified by qRT-PCR. (**D–F**) Primary BMECs from Dhh[ECKO] and control mice were isolated and cultured on Lab-Tek. (**D**) CDH5 (in green), (**E**) ZO1 (in green), and (**F**) CLDN5 (in red) localizations were evaluated by immunofluorescent staining of a confluent cell monolayer. Nuclei were stained with DAPI (in blue). The experiment was repeated 3×. (**G–L**) CNS MECs were isolated from Dhh[ECKO] and control mice, seeded and cultured until confluency, and (**G, H**) CDH5, (**G, I**) ZO1, (**G, J**) FOXO1, (**G, K**) and p-FOXO1 expression were quantified by western blot. (**L**) BMECs were isolated from Dhh[ECKO] mice and control mice, seeded and cultured until confluency and starved for 24 h.

Control BMECs were then treated with DMSO and Dhh$^{ECKO}$ BMECs with DMSO versus an inhibitor of FOXO1 (AS1842856). **(L)** CLDN5 expression was then quantified by qRT-PCR and **(M)** CLDN5 (in red) localization was evaluated by immunofluorescent staining of a confluent cell monolayer in Dhh$^{ECKO}$ BMECs treated with DMSO versus an inhibitor of FOXO1 (AS1842856). Nuclei were stained with DAPI (in blue). The experiment was repeated 3×. **(N)** Summary outline of the up-regulation of endothelial junctions by the morphogen DHH through inhibition of FOXO1 transcriptional activity. $^*P \leq 0.05$, $^{**}P \leq 0.01$, $^{***}P \leq 0.001$ Mann–Whitney U test. $^*P \leq 0.05$ Kruskal–Wallis test. The underlying data for Fig 2 can be found in S1 Data (https://doi.org/10.6084/m9.figshare.12625034.v6). $^*$It is important to note that CNS endothelial cells are from a pooled source including both brain and spinal cord tissues. Therefore, the resulting cell cultures/lysates may be heterogeneous in their use of DHH. This remark applies to Figs 1–4. AKT, serine/threonine kinase 1; BMEC, brain microvascular endothelial cell; CDH5, cadherin5; Cldn5, claudin5; CNS, central nervous system; CTNNβ1, catenin beta 1; Ctrl, control; DHH desert hedgehog; FOXO1, forkhead box O1; MEC, microvascular endothelial cells; NS, non-significant; PTCH1, Patched-1; qRT-PCR, quantitative reverse transcription polymerase chain reaction; ZO1, zonula occludens 1.

CTNNβ1 to inhibit transcription factor forkhead box O1 (FOXO1) [27] via PI(3)K–AKT-dependent phosphorylation, which thereby up-regulates the expression of the endothelial tight junction protein CLDN5 [27]. Therefore, we next measured the expression level of the phosphorylated form of FOXO1 (p-FOXO1) in CNS MEC cultures from Dhh$^{ECKO}$ and control (Dhh$^{Flox/Flox}$) mice and demonstrated that p-FOXO1 is down-regulated in Dhh$^{ECKO}$ mice compared to controls (Fig 2G, 2J–2K). We then treated CNS MEC cultures from Dhh$^{ECKO}$ mice with a cell permeable inhibitor of the transcription factor FOXO1 (AS1842856), which blocks the transcription activity of FOXO1, and measured *Cldn5* mRNA expression. We demonstrated that *Cldn5* mRNA expression, in Dhh$^{ECKO}$ CNS MEC cultures treated with the inhibitor of FOXO1, returns to the expression level of control CNS MEC cultures, unlike the Dhh$^{ECKO}$ CNS MEC cultures treated with DMSO (Fig 2L and S1 Data). Additionally, in Dhh$^{ECKO}$ CNS MEC cultures treated with the inhibitor of FOXO1, CLDN5 displays a well-defined pattern of sharp contours at endothelial cell–cell contacts. In contrast, in Dhh$^{ECKO}$ CNS MEC cultures treated with DMSO, a broader, more irregular pattern is detected at endothelial cell–cell contacts (Fig 2M).

We concluded that endothelial autocrine DHH expression at the BBB maintains the pool of CDH5-CTNNβ1 signaling in endothelial cells, which promotes *Cldn5* mRNA expression and maintains a well-defined CLDN5 pattern of sharp contours at endothelial cell–cell contacts through the inhibitory phosphorylation of the transcription factor FOXO1 (Fig 2N).

## In the white matter, endothelial-specific Dhh inactivation induces BBB permeability associated with endothelial and astrocytic activation in vivo

In vivo, on spinal cord sections from control and Dhh$^{ECKO}$ mice, we confirmed that the expression of adherens junction CDH5 and tight junction CLDN5, when normalized for the number and length of blood vessels, is down-regulated under resting conditions (Fig 3A–3D, S1 Data and S3A Fig) and demonstrated that it is associated with an increase accumulation of serum proteins (fibrinogen (FGB) and albumin (ALB)) [28] (Fig 3B, 3E, 3F, S1 Data, S3B and S3C Fig) around the vessels, suggesting BBB opening.

We then analyzed the activation status of both the spinal cord endothelium and Glia Limitans in Dhh$^{ECKO}$ and control littermates. We chose ICAM1 as a marker of endothelial activation and GFAP as a marker of astrocyte reactivity because they are both widely published in the context of inflamed CNS tissues and represent a strong indicator of a reactive response of the CNS endothelium and Glia Limitans.

Using spinal cord sections, we revealed that ICAM1 is up-regulated at the endothelium in Dhh$^{ECKO}$ mice compared to littermate controls (Fig 3G, 3I and S1 Data) and associated with a regionalized up-regulation of GFAP, a marker of astrocyte activation, in white matter (Fig 3H, 3J and S1 Data) but not gray mater (Fig 3H, 3K and S1 Data).

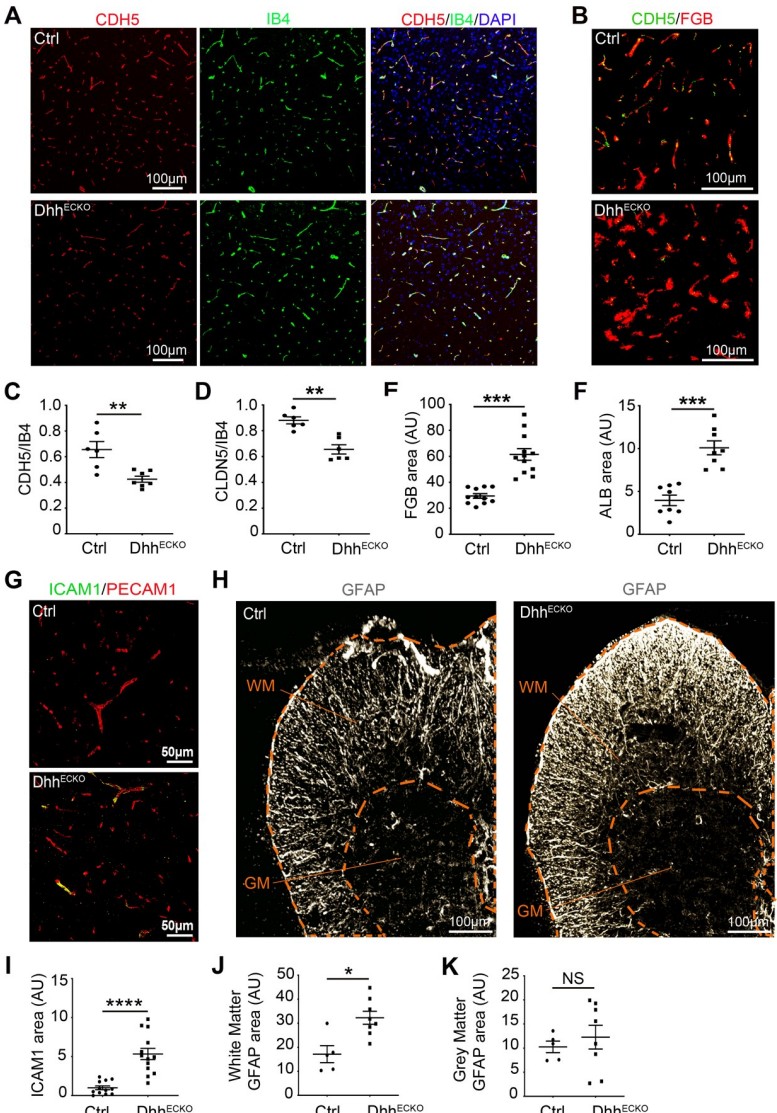

**Fig 3. Endothelial-specific *Dhh* inactivation induces BBB permeability associated with endothelial and astrocytic activition in vivo: (A–F)** Spinal cord sections were harvested from Dhh$^{ECKO}$ mice and littermate controls and immunostained with anti-IB4 (in green), anti-CDH5 (in red or green), anti-CLDN5 (in red), anti-FGB (in red), and anti-ALB (in red) antibodies. **(A)** Representative IB4/CDH5 and **(B)** CDH5/FGB staining are shown. **(C)** CDH5/IB4, **(D)** CLDN5/IB4, **(E)** FGB, and **(F)** ALB positive areas were quantified (Dhh$^{ECKO}$ *n* = 6 to 12, WT *n* = 6 to 11). **(G–K)** Spinal cord sections were harvested from Dhh$^{ECKO}$ mice and littermate controls and immunostained with anti PECAM1 (in red), anti-ICAM1 (in green), and anti-GFAP (in gray) antibodies. **(G, H)** Representative PECAM1, ICAM1, and GFAP staining are shown. **(H)** GFAP staining regionalization (WM and GM) is highlighted by orange dotted lines on the images. **(I)** ICAM1 and **(J, K)** GFAP positive areas in the WM **(J)** and GM **(K)** were quantified (Dhh$^{ECKO}$ *n* = 8 to 13, WT *n* = 5 to 12). $^*P \leq 0.05$, $^{**}P \leq 0.01$, $^{***}P \leq 0.001$, $^{****}P \leq 0.0001$ Mann–Whitney U test. The underlying data for Fig 3 can be found in S1 Data (https://doi.org/10.6084/m9.figshare.12625034.v6). *It is important to note that CNS endothelial cells are from a pooled source including both brain and spinal cord tissues. Therefore, the resulting cell cultures/lysates may be heterogeneous in their use of DHH. This remark applies to Figs 1–4. ALB, albumin; AU, arbitrary units; BBB, blood–brain barrier; CDH5, cadherin5; Cldn5, claudin5; Ctrl, control; DHH desert hedgehog; FGB, fibrinogen; GFAP, glial fibrillary acidic protein; GM, gray matter; IB4, isolectin B4; ICAM1, intercellular adhesion molecule 1; NS, non-significant; PECAM1, platelet/endothelial cell adhesion molecule 1; WM, white matter; WT, wild type.

Here, we detected that, in the white matter, DHH expression at the endothelium controls BBB tightness and demonstrated that endothelial-specific Dhh inactivation at the BBB drives endothelial and astrocyte activation.

## Dhh$^{ECKO}$-induced BBB breakdown is sufficient to induce a secondary CNS protective barrier at the Glia Limitans

As we already demonstrated in Fig 3, Dhh$^{ECKO}$ mice display BBB leakage, whereas control littermates feature a tight BBB.

Although we demonstrated BBB leakage in Dhh$^{ECKO}$ mice (Fig 3B, 3E, 3F and S1 Data), we noticed that infiltrating plasmatic proteins are concentrated around the vascular area in arterioles and venules and not seamlessly distributed within the parenchyma**. To verify this observation, we quantified the distribution of immunoglobulin G (IgG) and a smaller sized dye (70 kDa fluorescein isothiocyanate (FITC) Dextran) in the 3 compartments (lumen, PVS, and parenchyma). Specifically, in control mice, there is no significant endothelial permeability, with more than 95% of IgG and 97% of 70 kDa FITC Dextran contained in the lumen of blood vessels and 5% of IgG and 3% of 70 kDa FITC Dextran segregated in the PVS area limited by the astrocytic end feet (aquaporin 4 (AQP4) or laminin (LAM) antigen) on 1 side and the vessel wall (PECAM1 or LAM antigen) on the other side, and none found in the parenchyma (S1 Data, S4A–S4E Fig and S2 Data). The quantification protocol of IgG and 70 kDa FITC Dextran distribution within the lumen, PVS, and parenchyma is described in S4 Fig. In Dhh$^{ECKO}$ mice, vascular leakage is significant, but strikingly, 50% of IgG and 50% of 70 kDa FITC Dextran is contained into the PVS, while none is found in the parenchyma, indicating the presence of a secondary barrier at the Glia Limitans (S1 Data, S4A–S4E Fig and S2 Data). Interestingly, some overlap is observed between the IgG signal and the AQP4 signal, concentrated within the internal surface of the Glia Limitans (Fig 4A). Thus might reflect a potential interaction between the inner face of astrocyte end feet and IgG accumulated in the PVS, as astrocytes express Fc receptors (cell surface receptors for IgG), which play a role in both CNS health and disease [29,30]. The perivascular trapping of IgG in Dhh$^{ECKO}$ mouse CNS was confirmed using co-immunostaining of IgG and LAM antigen, which mark the endothelial and astrocytic basement membranes (S5A and S5B Fig).

We have previously found that reactive astrocytes express tight junctions, notably CLDN4, under inflammatory conditions in a mouse model of multiple sclerosis (EAE) [26]. Here, we found that these data are relevant to human disease since CLDN4 is also expressed by reactive astrocytes, with stronger CLDN4 labeling intensity at the Glia Limitans, in active cortical lesions from multiple sclerosis patients (Fig 4D). Based on the above results, we investigated whether the PVS entrapment of plasmatic proteins observed in Dhh$^{ECKO}$ mice is linked to the expression of the tight junction CLDN4 at the Glia Limitans in response to BBB permeability (Fig 4E–4H and S1 Data). We showed that CLDN4 is expressed at the Glia Limitans in Dhh$^{ECKO}$ mice but not control littermates (Fig 4E–4H and S1 Data) using isolated neurovascular enriched fractions. Small intestine samples were used as a positive control for the quantification of CLDN4 by western blot (S6A Fig).

Altogether, these results suggest that, in Dhh$^{ECKO}$ mice, spontaneous BBB permeability leads to the establishment of a physical barrier at the Glia Limitans, characterized by the expression of the tight junction protein CLDN4. Therefore, in Dhh$^{ECKO}$ mice, astrocytic end feet at the Glia Limitans are "preconditioned" to form a secondary barrier protecting the parenchyma. (**It is important to note that this study focuses on arterioles and venules but not capillaries. Indeed, CLDN4 is only up-regulated in Dhh$^{ECKO}$-enriched neurovascular fractions, which are 100 μm and larger in diameter (Fig 4F–4H and S1 Data). In the lysates obtained with enriched

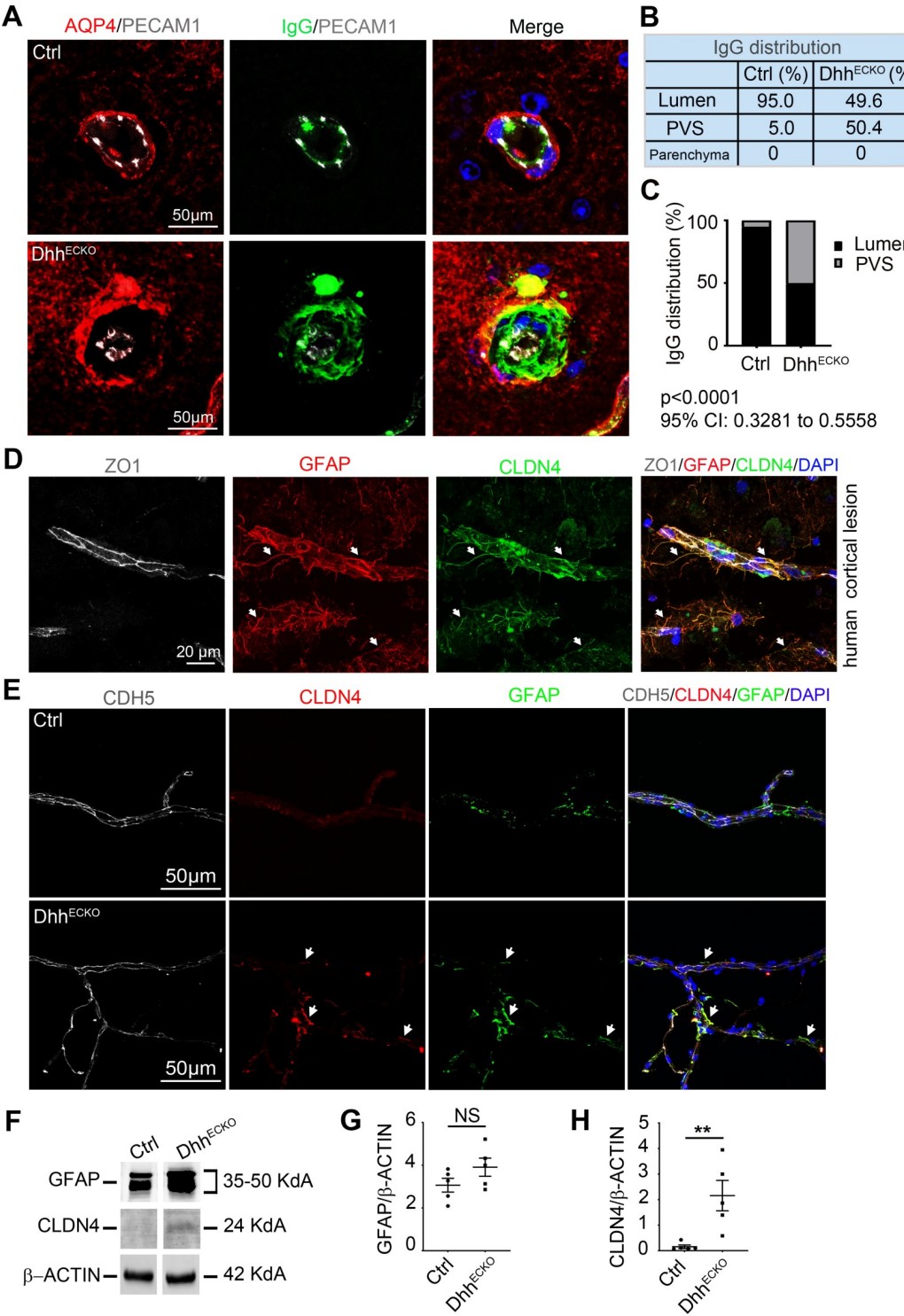

**Fig 4. Dhh^ECKO-induced BBB breakdown is sufficient to induce a secondary CNS protective barrier at the Glia Limitans. (A–C)** Spinal cord sections were harvested from Dhh^ECKO mice and littermate controls and **(A)** immunostained with anti-PECAM1 (in gray), anti-AQP4 (in green), and anti-IgG (in red) antibodies (nuclei were stained with DAPI (in blue)), and **(B, C)** the distribution of IgG within the lumen, PVS, and parenchyma was quantified. We quantified the amount of IgG in the PVS and parenchyma and inferred from it the amount of IgG contained in the lumen. (Dhh^ECKO n = 6, WT n = 6). P < 0.0001, 95% CI: 0.3281 to 0.5558, chi-squared test. **(D)** Human active multiple sclerosis cortical lesions were obtained from the NeuroCEB biobank and immunostained with anti-ZO1 (in gray), anti-GFAP (in red), and

anti-CLDN4 (in green) antibodies. Strong CLDN4 signal at the Glia Limitans is indicated by white arrows. Nuclei were stained with DAPI (in blue). **(E–H)** Enriched neurovascular fractions were isolated from Dhh$^{ECKO}$ and control mouse CNS and **(E)** immunostained with anti-CDH5 (in gray), anti-CLDN4 (in red), and anti-GFAP (in green) antibodies (nuclei were stained with DAPI (in blue)). **(F)** Representative blots of GFAP and CLDN4 expression levels on Dhh$^{ECKO}$ and control mouse enriched neurovascular fractions were shown, and **(G, H)** GFAP and CLDN4 expression levels were quantified by western blot of Dhh$^{ECKO}$ and control mouse enriched neurovascular fractions. (Dhh$^{ECKO}$ $n = 5$, WT $n = 5$). **$P \leq 0.01$, Mann–Whitney U test. The underlying data for Fig 4 can be found in S1 Data (https://doi.org/10.6084/m9.figshare.12625034.v6). *It is important to note that CNS endothelial cells are from a pooled source including both brain and spinal cord tissues. Therefore, the resulting cell cultures/lysates may be heterogeneous in their use of DHH. This remark applies to Figs 1–4. AQP4, aquaporin 4; BBB, blood–brain barrier; CDH5, cadherin5; CI, confidence interval; CLDN4, Claudin4; CNS, central nervous system; Ctrl, control; DHH desert hedgehog; GFAP, glial fibrillary acidic protein; IgG, immunoglobulin G; NS, non-significant; PECAM1, platelet/endothelial cell adhesion molecule 1; PVS, perivascular space; WT, wild type; ZO1, zonula occludens 1.

neurovascular fractions, which are 20 μm and larger, CLDN4 expression level is not different between the Dhh$^{ECKO}$ and control group (S6B–S6D Fig and S2 Data). In this condition, CLDN4 expression in arterioles and venules might have been lost in the crowd of capillaries.)

## Endothelial signals can drive astrocyte barrier properties at the Glia Limitans

Given the above results, we wanted to determine if astrocyte barrier formation requires signals from the endothelial BBB or from the plasmatic protein perivascular infiltrate. To do so, we first studied in vitro the response of normal human astrocytes (NHAs) to HBMEC conditioned media versus plasmatic proteins from healthy donors. HBMECs used to produce the conditioned media were treated with either the osmotic agent Mannitol or the pro-permeability factor vascular endothelial growth factor A (VEGFA) to induce BBB breakdown (S7A Fig and S2 Data) through various methods.

We demonstrated that *Gfap* (Fig 5A, 5E–5I and S1 Data), *aldehyde dehydrogenase 1 family, member l1 (Aldh1l1)* (Fig 5B and S1 Data) (markers of astrocyte reactivity), and *Cldn4* mRNA expression (Fig 5D, 5A–5H, 5J and S1 Data) are up-regulated in the NHA treated with HBMEC-conditioned media but not in the NHA treated with plasma from healthy donors. *vimentin (Vim)* (marker of astrocyte reactivity) mRNA expression level was not modulated in any condition (Fig 5C and S1 Data).

To confirm this observation in vivo, we delivered murine VEGFA or murine plasmatic proteins into the left cerebral cortex of adult mice and evaluated the consequences on CLDN4 expression by astrocytes. PBS stereotactic administration was used as a control. Importantly, VEGFA cortical stereotactic injection has already been shown to efficiently induce BBB breakdown in mice [17,31].

In vivo, GFAP (Fig 5K–5N and S1 Data) and CLDN4 (Fig 5K–5M, 5O and S1 Data) are induced in mouse cortex after murine VEGFA and murine plasmatic protein treatments, with VEGFA having a much stronger effect than plasmatic proteins.

We concluded that permeable endothelial monolayers produce signals that can drive astrocyte reactivity and tight junction expression. Plasmatic protein involvement in controlling astrocyte barrier behavior is, however, less clear as astrocyte reactivity and tight junction expression are up-regulated in vivo but not in vitro when treated with plasma; further investigations will be necessary to identify the mechanisms involved.

## Mice with endothelial Dhh inactivation display reduced disability in a model of multiple sclerosis during the onset of the disease

To examine the impact of these findings on disease severity, we investigated the phenotype of induced experimental multiple sclerosis (EAE) in Dhh$^{ECKO}$ and control mice.

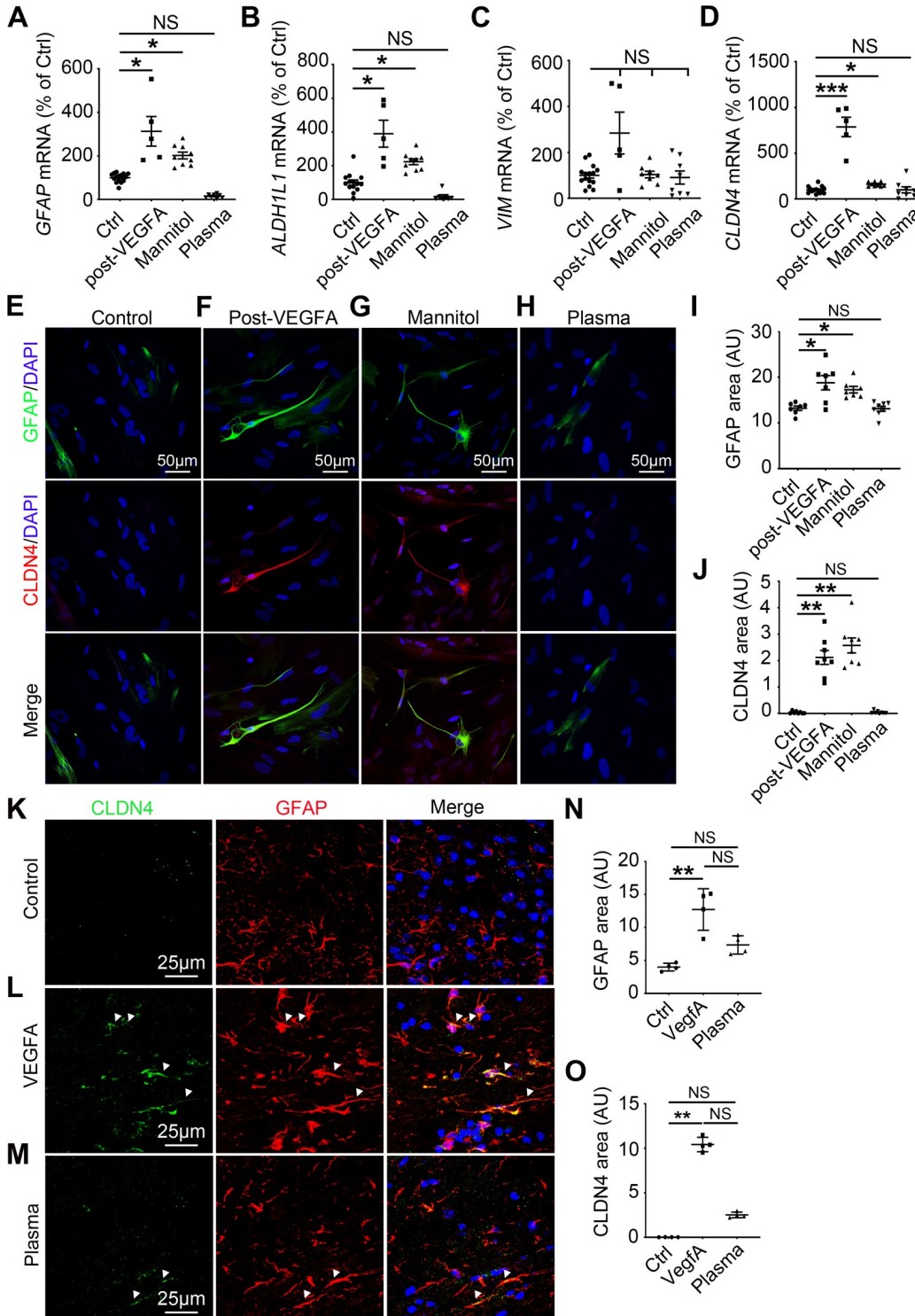

**Fig 5. Endothelial signals can drive astrocyte barrier properties at the Glia Limitans. (A–K)** NHA were cultured until confluency and starved for 24 h. NHA were then treated for 24 h with HBMEC medium from untreated cells (control condition), conditioned media from HBMECs treated with VEGFA, conditioned media from HBMECs treated with Mannitol, or HBMEC medium with 20% plasma from healthy donors (Mannitol and VEGFA were washed out of the HBMEC cultures before the medium was used to treat the NHA cultures). **(A)** *Gfap*, **(B)** *Aldh1l1*, **(C)** *Vim*, and **(D)** *Cldn4* mRNA expression was quantified by qRT-PCR. (Post-VEGFA $n = 5$, Mannitol $n = 8$ to 9, Plasma $n = 7$ to 8, Vehicle control $n = 14$ to 15). **(E–H)** GFAP (in green) and **(E)** CLDN4 (in red) localizations were evaluated by immunofluorescent

staining of a confluent NHA monolayer. Nuclei were stained with DAPI (in blue). The experiment was repeated 3 times. **(I, J)** GFAP and CLDN4 positive areas were then quantified. (Post-VEGFA $n$ = 7 to 8, Mannitol $n$ = 7 to 8, Plasma $n$ = 8, Vehicle control $n$ = 7 to 8). **(K–O)** Cerebral cortices of 10-week-old C57BL/6 mice were harvested 24 h following stereotactic microinjection of murine VEGFA (60 ng in 3 μL PBS), healthy C57BL/6 mouse plasma (3 μL), or vehicle control (3 μL PBS). **(K–M)** Cortical lesions were immunostained with anti-GFAP and anti-CLDN4 antibodies. **(N)** GFAP positive areas and **(O)** CLDN4 positive areas were quantified (VEGFA $n$ = 4, healthy C57BL/6 mouse plasma $n$ = 4, Vehicle control $n$ = 4). $^*P \le 0.05$, $^{**}P \le 0.01$, $^{***}P \le 0.001$ Kruskal–Wallis test. The underlying data for Fig 5 can be found in S1 Data (https://doi.org/10.6084/m9.figshare.12625034.v6). Aldh1l1, aldehyde dehydrogenase 1 family, member l1; AU, arbitrary units; CLDN4, Claudin4; Ctrl, control; GFAP, glial fibrillary acidic protein; HBMEC, human brain microvascular endothelial cells; NHA, normal human astrocytes; NS, non-significant; qRT-PCR, quantitative reverse transcription polymerase chain reaction; VEGFA, vascular endothelial growth factor A; Vim, vimentin.

We observed that in control mice, neurologic deficits were observed from day 9 and increased in severity until day 18, when clinical score stabilized at a mean of 3.2, representing hind limb paralysis. In contrast, the onset of clinical signs in Dhh^ECKO mice was first seen 4 days later, and the clinical course was much milder. In Dhh^ECKO mice, disease reached a plateau at day 21 at a mean of 2.3, indicating hind limb weakness and unsteady gait, a mild phenotype (Fig 6A and S1 Data). The EAE peak score (Fig 6B and S1 Data) and average score during the time of disability (Fig 6C and S1 Data) were both decreased in Dhh^ECKO mice, but there were no significant changes in survival and mortality rates (Fig 6D and S1 Data).

Interestingly, the reduced clinical course of EAE in Dhh^ECKO mice was much more marked during the onset of the disease (between day 12 and day 20 post EAE induction). However, when the plateau phase was reached (after day 21 post EAE induction), the clinical score difference between the Dhh^ECKO and the control group was greatly reduced and coincided with an acceleration of the mortality rate in both groups (Fig 6A–6D and S1 Data). One explanation for this observation is that, in the Dhh^ECKO group, inflammatory cells, accumulated in the PVS, end up degrading astrocytic tight junctions by secreting proteases. This previously described phenomenon [26] explains how inflammatory cells trapped in the PVS can eventually pass through the astrocytic barrier at the Glia Limitans and thereby enter the CNS, thus inactivating Dhh at the BBB. This in turn slows disease progression until the perivascular accumulation of inflammatory cells causes the degradation of the astrocyte secondary barrier, leading to the deterioration of the clinical course of EAE in the Dhh^ECKO mice.

Importantly, the clinical course in the Dhh^ECKO mice is correlated with strikingly decreased areas of demyelination as compared to the control cohort (Fig 6E, 6F and S1 Data). Critically, these studies reveal that the clinical course and pathology of EAE are strongly reduced in Dhh^ECKO mice during the onset of the disease.

We concluded that endothelial Dhh knockdown-induced BBB opening is associated with a clinical protective effect during the onset of the disease in a model of multiple sclerosis.

## Mice with endothelial Dhh inactivation display a reinforced barrier at the Glia Limitans, restraining access to the parenchyma to inflammatory infiltrate in a model of multiple sclerosis

Although Dhh^ECKO mice display equivalent FGB densities (Fig 7A and S1 Data) as well as numbers of CD45+ leukocytes (Fig 7B and S1 Data) in lesions compared to those in control mice, neuropathology in both cohorts appeared very different.

We found that while the BBB is permeable in both groups, with plasmatic protein extravasation associated with equivalent CDH5 densities, astrocyte reactivity in EAE lesions in Dhh^ECKO mice is greatly increased with GFAP immunoreactivity strongest at the Glia Limitans (Fig 7C, 7E and S1 Data). Moreover, infiltrating plasmatic proteins in Dhh^ECKO mice show less CNS parenchymal dispersion (Fig 7E, 7F and S1 Data), with 68.0% of FGB trapped into the

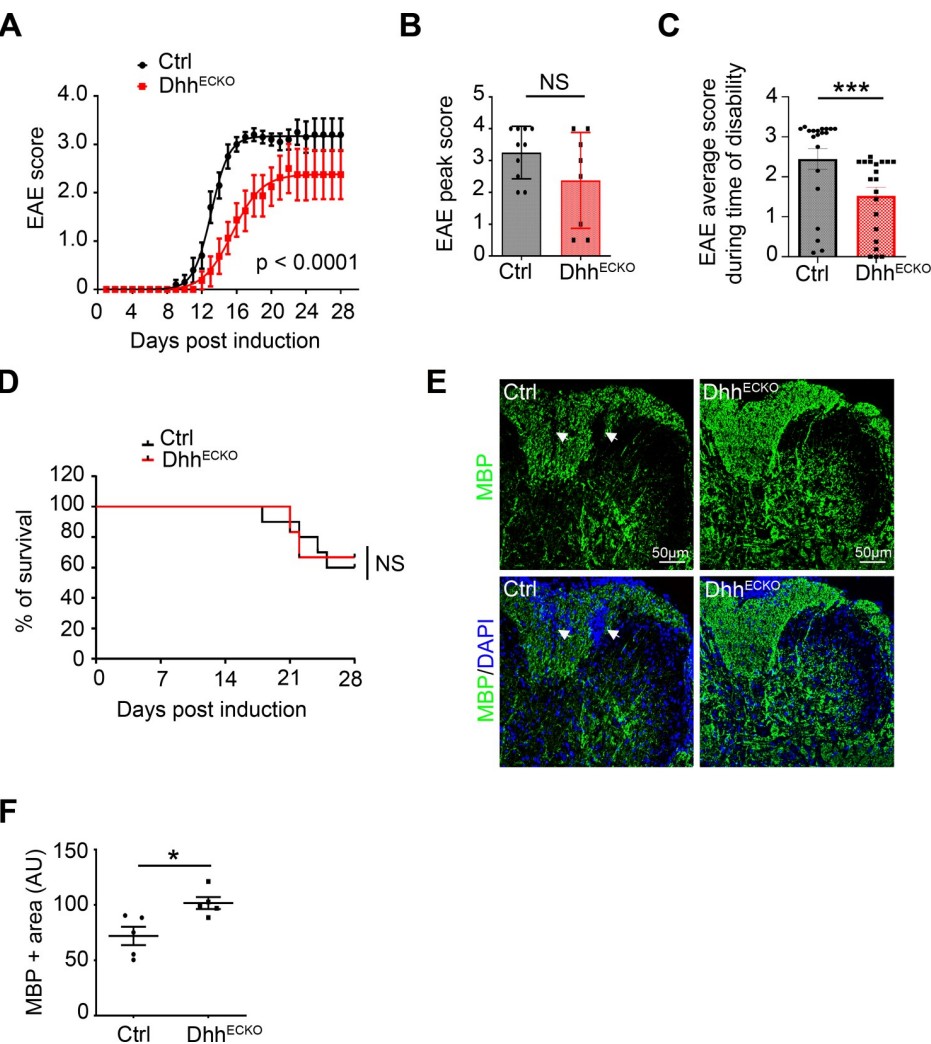

**Fig 6. Dhh<sup>ECKO</sup> mice display reduced disability in a model of multiple sclerosis during the onset of the disease.**
**(A)** Dhh<sup>ECKO</sup> and control mice induced with EAE were scored daily on a standard 5-point scale, nonlinear regression (Boltzmann sigmoidal). (Dhh<sup>ECKO</sup> *n* = 8, WT *n* = 10). **(B)** Dhh<sup>ECKO</sup> and control mice EAE peak score, **(C)** EAE average score during time of disability, and **(D)** mortality rate were quantified other the course of the disease. **(E, F)** Spinal cord EAE lesions from Dhh<sup>ECKO</sup> mice and littermate controls were harvested at 28 days post induction or at the time of euthanasia. **(E)** Dhh<sup>ECKO</sup> and control lesions were immunostained with an anti-MBP (in green) antibody; arrows indicate white matter loss areas. Nuclei were stained with DAPI (in blue). **(F)** MBP positive areas were quantified ((Dhh<sup>ECKO</sup> *n* = 5, WT *n* = 5). $^*P \leq 0.01$, Mann–Whitney U test. The underlying data for Fig 6 can be found in S1 Data (https://doi.org/10.6084/m9.figshare.12625034.v6). AU, arbitrary units; Ctrl, control; DHH desert hedgehog; EAE, experimental autoimmune encephalomyelitis; MBP, myelin basic protein; NS, non-significant; WT, wild type.

Glia Limitans in the Dhh<sup>ECKO</sup> cohort versus 32% in the control cohort (Fig 7F, S7B Fig and S1 Data).

In previous work from our laboratory [8], we showed that endothelial-specific deletion of Dhh in the peripheral vasculature is associated with vascular permeability and endothelial activation, notably in the lung, and that lipopolysaccharide (LPS) injection increased pulmonary neutrophil infiltration in Dhh<sup>ECKO</sup> mice compared to control littermates. Therefore, we can hypothesize that endothelial Dhh knockdown increases the peripheral recruitment and activation of inflammatory cells that need to travel to the CNS. However, in our study, we do not

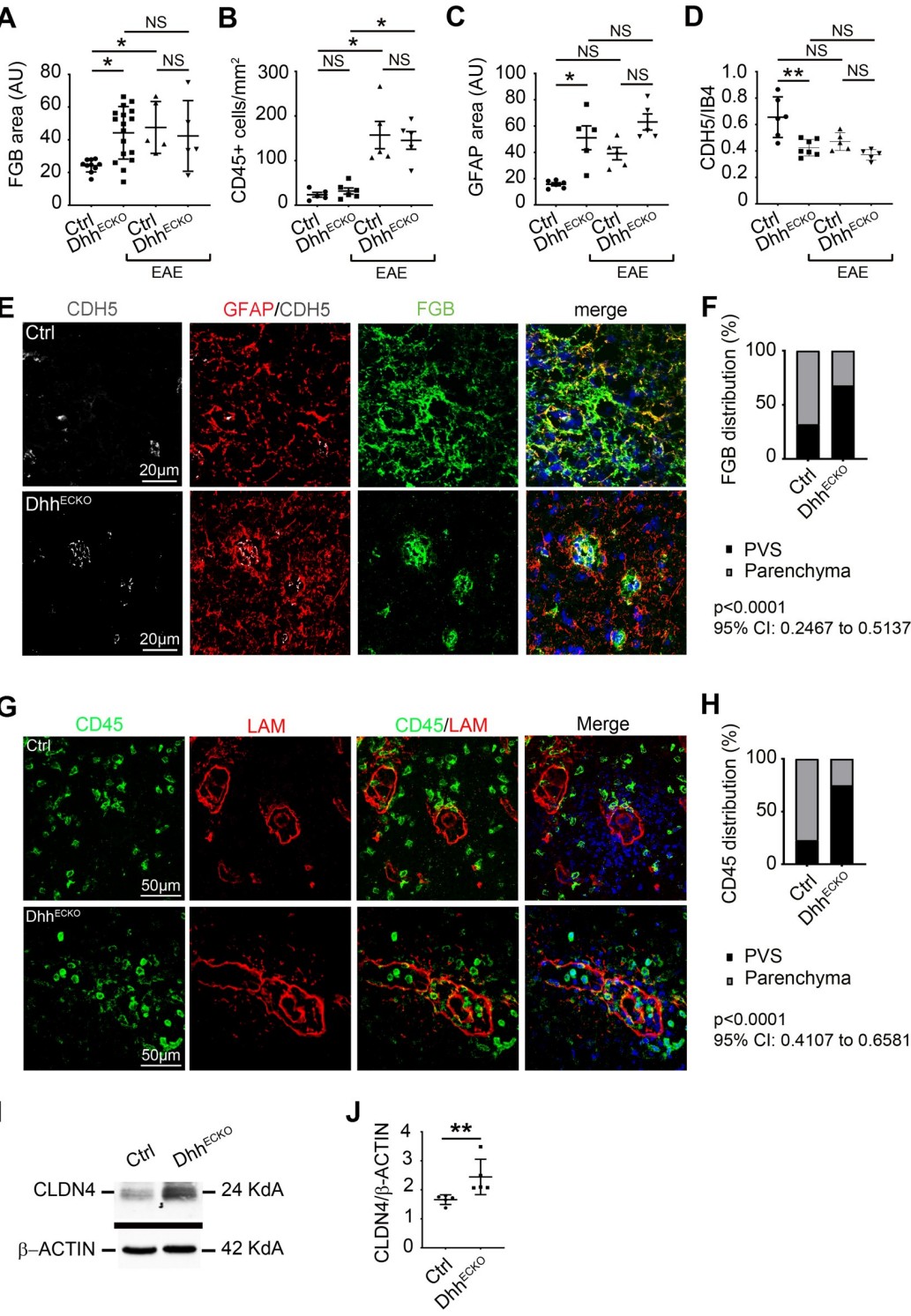

**Fig 7. Mice with endothelial Dhh knockdown display a reinforced barrier at the Glia Limitans restraining access to the parenchyma to inflammation in a model of multiple sclerosis. (A–H)** Spinal cord from Dhh[ECKO] mice and littermate controls under resting condition and after EAE induction were harvested. **(A–D)** Dhh[ECKO] and control sections were immunostained with anti-FGB, anti-CD45, anti-GFAP, anti-IB4, and anti-CDH5 antibodies. (A) FGB positive areas, (B) the number of CD45+ T cells per mm$^2$, (C) GFAP positive areas, and (D) CDH5/IB4 positive area were quantified (Dhh[ECKO] $n$ = 5 to 17, WT $n$ = 5 to 10). (E) Dhh[ECKO] and control EAE lesions were immunostained with anti-CDH5 (in gray), anti-FGB (in green), and anti-GFAP (in red) antibodies (Nuclei were stained with DAPI (in blue)), and (F) the

distribution of FGB within the Glia Limitans and the parenchyma was quantified. (Dhh$^{ECKO}$ $n = 6$, WT $n = 5$). $P < 0.0001$, 95% CI: 0.2467 to 0.5137, chi-squared test. (G) Dhh$^{ECKO}$ and control lesions were immunostained with anti-LAM (in red) and anti-CD45 (in green) antibodies (Nuclei were stained with DAPI (in blue)), and (H) the distribution of CD45+ T lymphocytes within the Glia Limitans and the parenchyma was quantified. $P < 0.0001$, 95% CI: 0.4107 to 0.6581, chi-squared test. (I) Representative blots of CLDN4 expression level on Dhh$^{ECKO}$ and control spinal cord EAE lesion lysates were shown, and (J) CLDN4 expression level was quantified by western blot on Dhh$^{ECKO}$ and control spinal cord EAE lesion lysates. (Dhh$^{ECKO}$ $n = 5$, WT $n = 5$). **$P \leq 0.01$, Mann–Whitney U test. *$P \leq 0.05$, **$P \leq 0.01$, Kruskal–Wallis test. The underlying data for Fig 7 can be found in S1 Data (https://doi.org/10.6084/m9.figshare.12625034.v6). AU, arbitrary units; CD45, cluster of differentiation 45; CDH5, cadherin 5; CI, confidence interval; CLDN4, Claudin4; Ctrl, control; DHH desert hedgehog; EAE, experimental autoimmune encephalomyelitis; FGB, fibrinogen; GFAP, glial fibrillary acidic protein; IB4, isolectin B4; LAM, laminin; NS, non-significant; PVS, perivascular space; WT, wild type.

observe any difference in terms of CD45+ leukocyte populations in the spinal cord of Dhh$^{ECKO}$ and control mice induced with EAE MOG$_{35\text{-}55}$. What we do observe is a significant difference in terms of the repartition of these CD45+ cell populations between both groups, with 77.1% of CD45+ cells trapped into the PVS in the Dhh$^{ECKO}$ cohort versus 25.1% in the control cohort (Fig 7G, 7H, S1 Data and S7C Fig).

Altogether, these data suggest less access through the Glia Limitans in the Dhh$^{ECKO}$ mice compared to the littermate control mice. Finally, we demonstrated that CLDN4 expression in spinal cord EAE lesion lysates is up-regulated in Dhh$^{ECKO}$ mice as compared to control mice (Fig 7I, 7J and S1 Data).

Collectively, data from Fig 4 to Fig 7 reveal that conditional loss of a key structural component of endothelial integrity at the BBB in Dhh$^{ECKO}$ mice leads to increased astrocyte reactivity and implementation of barrier properties at the Glia Limitans, allowing for less diffusion of plasmatic proteins and immune cells into the CNS parenchyma than in control mice. Therefore, in Dhh$^{ECKO}$ mice, astrocytic end feet at the Glia Limitans are "preconditioned" to form a barrier, explaining their ability to protect the parenchyma more efficiently during neuropathology than in controls, leading to the protective effect observed clinically. Thus, we identify BBB leakage, induced by the down-regulation of Dhh endothelial expression, as an important mechanism controlling Glia Limitans reactivity and barrier properties, and subsequently, tissue damage and clinical deficits in a model of human disease (Fig 8).

## Discussion

While it is now well established that BBB breakdown leads to soluble factor and inflammatory cell infiltration into the PVS during neuropathology [4], the function of the Glia Limitans barrier is just starting to be unraveled [15,20,24,26]. In the present study, we have enabled a different perspective on CNS barrier organization, unveiling the existence of 2 independent, dissociable states of the astrocyte and endothelial barriers in the neurovascular unit. Indeed, we confirmed that, just like the BBB, the Glia Limitans can form a protective barrier. For the first time, to our knowledge, we have demonstrated that BBB breakdown is sufficient to induce chronic barrier properties at the Glia Limitans, and we uncovered crosstalk from endothelial cells to astrocytes that restricts access to the parenchyma of plasmatic proteins and inflammatory cells during multiple sclerosis. Moreover, we showed that in Dhh$^{ECKO}$ mice, which display an open BBB, astrocytes express the tight junction protein CLDN4 under resting conditions. Therefore, under neuroinflammatory conditions, Glia Limitans in Dhh$^{ECKO}$ mice is primed with stronger barrier properties protecting against the onset and severity of EAE symptoms in the Dhh$^{ECKO}$ mice compared to control littermates.

However, 2 questions remain regarding barrier properties at the Glia Limitans: (1) Is this a specific aspect of the Dhh$^{ECKO}$ mouse model or more broadly applicable to any model of BBB breakdown; and (2) is it observed in other CNS pathologies than inflammation?

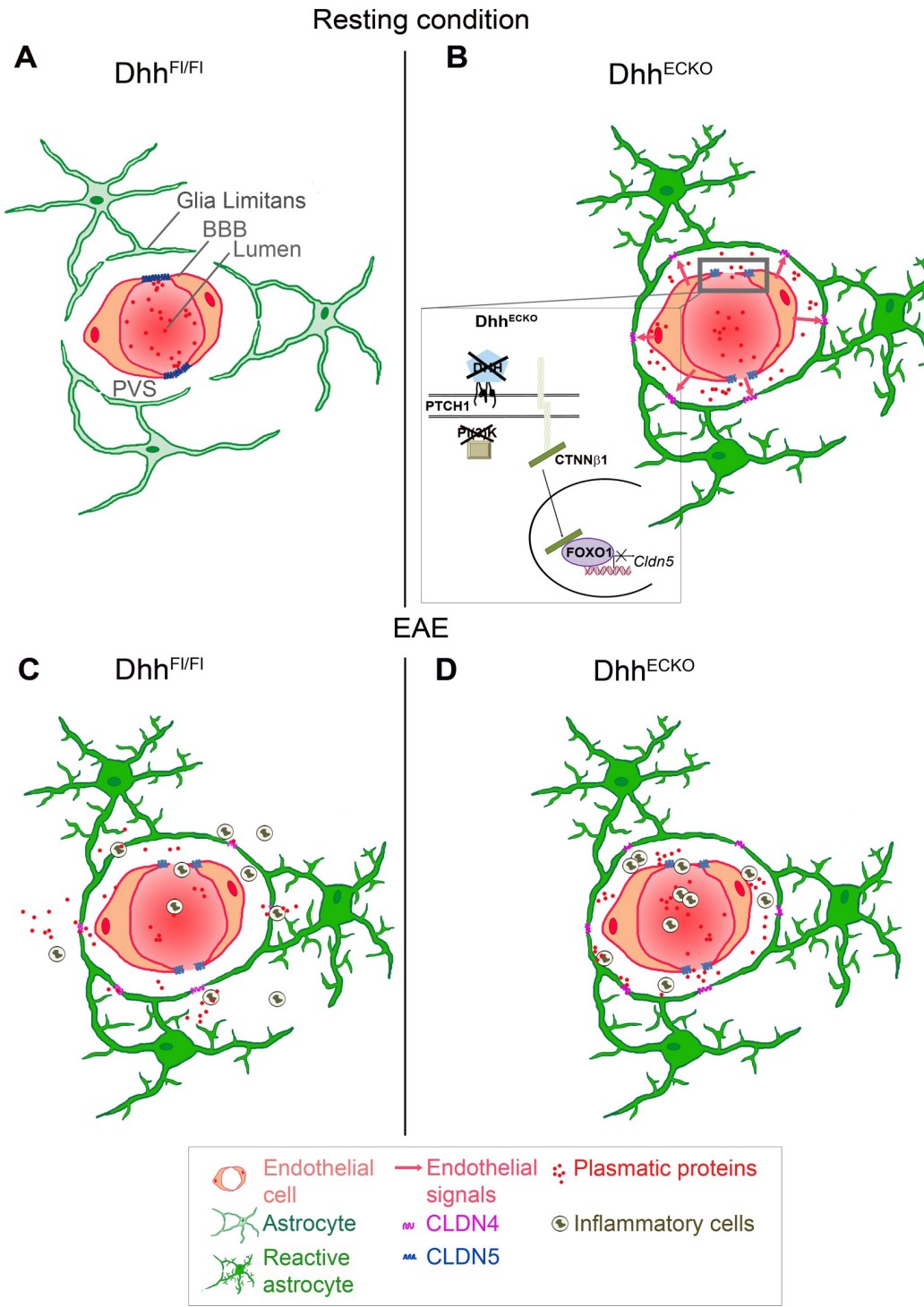

**Fig 8. Schematic of the BBB and Glia Limitans in Dhh^ECKO versus control mice, in health and inflammatory disease.**
Under resting conditions, in control mice, ECs express CLDN5 and CDH5, which reinforce a closed BBB. In Dhh^ECKO mice, CLDN5 and CDH5 are disrupted leading to BBB permeability; in turn, astrocytes of the Glia Limitans up-regulate CLDN4, closing the Glia Limitans and restricting incoming plasmatic proteins to the PVS. Under inflammatory condition, in both control and Dhh^ECKO mice, CLDN5 and CDH5 are disrupted leading to BBB permeability, astrocyte reactivity, and up-regulation of CLDN4. However, in Dhh^ECKO mice, astrocytic end feet of the Glia Limitans have been "preconditioned" to form a barrier, explaining their ability to trap plasmatic proteins and inflammatory cells in the PVS, and thus protecting the parenchyma more efficiently than in the controls. BBB, blood–brain barrier; CDH5, cadherin 5; CLDN4, Claudin4; CLDN5, claudin5; CTNNβ1, catenin β1; DHH desert hedgehog; EC, endothelial cells; FOXO1, forkhead box O1; PTCH1, Patched-1; PVS, perivascular space.

A model of pericyte-deficient mice featuring BBB breakdown has been described; but, in this model, endothelial permeability is due to increased transcytosis and not junction degradation as in the Dhh$^{ECKO}$ mouse model. Therefore, it would be interesting to study the behavior of astrocytes in this model, notably their capacity to express CLDN4 and to restrain parenchymal access to plasmatic proteins. Should astrocyte barrier properties be observed in the pericyte-deficient mouse model, thus would be suggestive for a more generalized role of BBB breakdown in driving astrocyte barrier properties, either by parallel or convergent mechanisms.

The expression of CLDN4 by astrocytes has only been identified in mouse models of CNS inflammation (acute CNS inflammation model (stereotactic injection of the pro-inflammatory cytokine IL-1β and model of multiple sclerosis: EAE MOG$_{35-55}$) [26]. However, in the present study, we demonstrated that CLDN4 is expressed by astrocytes under resting conditions in response to BBB opening in the Dhh$^{ECKO}$ mouse model and after cortical stereotactic injection of the pro-permeability factor VEGFA. The next step would be to examine astrocytic CLDN4 expression in other pathologies of the CNS, notably Alzheimer's disease, stroke, or amyotrophic lateral sclerosis in which BBB permeability has been identified as a critical pathophysiological player. This could implicate CLDN4-mediated barrier function in astrocytes as a more generalized defense against BBB opening in other chronic diseases of the CNS.

The critical role of Hh signaling in CNS neuroinflammation was first highlighted in 2011; this study revealed that during EAE, the morphogen SHH is expressed by reactive astrocytes and participates in the maintenance of BBB integrity [4]. Following this discovery, our group found that DHH is physiologically expressed at the BBB in adults [8]. Here, we demonstrated for the first time that DHH is down-regulated at the BBB during EAE and that DHH knockdown is sufficient to induce BBB permeability by inhibiting CDH5 and CLDN5 expression through the modulation of FOXO1 activity, strengthening the idea that HH signaling is essential to control BBB integrity both physiologically and under multiple sclerosis conditions. Based on our results and the literature, we hypothesize that DHH is necessary to maintain BBB tightness under physiological conditions and that DHH down-regulation under inflammatory conditions might be offset by astrocytic SHH secretion to maintain BBB homeostasis during disease progression.

Over the past years, the field has begun to acknowledge the fact that the BBB is not the sole line of defense of the CNS and that the astrocytic end feet of the Glia Limitans might play a role in restricting access to the parenchyma. Indeed, it was first described that in spinal cord injury, astrocyte scar borders corral inflammatory cells within areas of damaged tissue [32,33]. Moreover, we have found that during EAE, reactive astrocytes of the Glia Limitans form tight junctions of their own containing CLDN4 [26], a junction protein also expressed in tightly sealed epithelia [34,35]. Noteworthy is the fact that down-regulation or reorganization of CLDNs and other tight junction proteins has been implicated in permeability in various tissues, particularly the gut [19,36,37]; however, reports of dynamic tight junction protein induction resulting in functional barrier formation have been rare [38,39]. Here, we have shown for the first time that genetically induced disruption of endothelial junctions is sufficient to induce CLDN4 expression at the Glia Limitans under resting condition, identifying an inducible astrocyte barrier mediated fully by signals transmitted by the open BBB.

It has already been described, by our group and others, that astrocytes can send signals, notably VEGFA [18], thymidine phosphorylase (TYMP) [17], and SHH [4], to the BBB to modulate its state (tight versus permeable). In this study, we identify a reciprocal signaling pathway demonstrating that BBB endothelial junction disruption leads to CLDN4 expression at the Glia Limitans. Interestingly, endothelial cell capacity to send signals to neighboring cells has been previously identified, notably in the context of pericyte (mural cells associated with arterioles, capillaries, and venules) recruitment at the vascular wall [40]. Specifically, it has

been shown that platelet-derived growth factor subunit b (PDGFB) is secreted from angiogenic sprout endothelium where it serves as an attractant for co-migrating pericytes, which in turn express platelet-derived growth factor receptor beta (PDGFRβ) [41]. Based on these arguments and our results, it appears highly likely that endothelial signals can be sent to astrocytes. Identifying such signals will be the aim of future studies by our group.

We then showed that in animals, which exhibit an open BBB (Dhh$^{ECKO}$ mice), astrocytic end feet of the Glia Limitans form a barrier more efficiently than in the littermate controls, leading to the protective effect observed clinically in the model of multiple sclerosis. This is somehow reminiscent of what is observed in brain ischemic preconditioning where a mild nonlethal ischemic episode (preconditioning) can produce resistance to a subsequent more severe ischemic insult [42,43]. Here, inducing BBB opening and PVS plasmatic protein accumulation produces resistance to the subsequent massive inflammatory infiltration induced by multiple sclerosis development. This could account for the infrequency of recurrent multiple sclerosis relapse/lesion formation at the same location in the CNS. Interestingly, among neurons and nonneuronal cells, astrocytes are considered increasingly important in regulating cerebral ischemic tolerance [44], and a parallel can be easily drawn between these results and ours showing a major role for "preconditioned" astrocytes in the control of "chronic neuroinflammation tolerance" and protection against further relapse.

In light of the above observations, we may assume that the CNS has the ability to protect itself against isolated BBB leakage episodes through a secondary barrier at the Glia Limitans that takes over once the BBB is open. Moreover, it suggests that manipulation of the BBB and Glia Limitans in combination may have greater potential than either alone to control CNS entry of leukocytes and pro-inflammatory soluble factors in conditions such as multiple sclerosis and perhaps more widely. Indeed, taking into account both components of the neurovascular unit is of translational interest, notably to limit CNS parenchymal access to pathogenic agents by strengthening the Glia Limitans once the BBB is open in cardiovascular diseases such as brain ischemic strokes [45], neuroinfections [46], and neurodegeneration (Parkinson/Alzheimer's diseases and vascular dementia) [47], or to facilitate parenchymal access to drugs, by opening the BBB and Glia Limitans together, in CNS tumor treatment [48]. Along similar lines, it is unknown how the barrier properties of the Glia Limitans may impact the pharmacokinetics of drugs that must enter the CNS parenchyma in conditions such as multiple sclerosis, which may account for treatment failure.

In summary, our study first demonstrates the critical role of DHH in maintaining BBB integrity. We find that DHH is down-regulated during the animal model of multiple sclerosis and that Dhh knockdown leads to BBB opening. Using Dhh knockdown as a tool to cause BBB opening, we then show that BBB permeability is sufficient to induce a secondary barrier at the Glia Limitans, mediated by CLDN4 and astrocyte reactivity. These findings not only highlight the capacity for bidirectional signaling between the endothelial BBB and the astrocytic Glia Limitans in modulating the double barriers of the CNS but also provide support for a novel concept of "chronic neuroinflammatory tolerance", in which chronic induction of Glia Limitans barrier properties by BBB opening may lead to a protective effect against neuroinflammatory disease activity and progression.

## Methods

### Human tissues

Cortical sections from multiple sclerosis patients (active lesions) and healthy controls (frontal cortex) were obtained from the NeuroCEB bio bank (https://www.neuroceb.org/fr). The sections were 30 μm thick and obtained from fresh frozen samples.

## Mice

*Dhh Floxed (Dhh^Flox)* mice were generated at the "Institut Clinique de la Souris" through the International Mouse Phenotyping Consortium (IMPC) from a vector generated by the European conditional mice mutagenesis program, EUCOMM, and described before [8].

The Cre recombinase in cadherin5 (*Cdh5*)-*Cre^ERT2* mice was activated by intraperitoneal injection of 1-mg tamoxifen (Sigma Aldrich, St. Louis, Missouri, United States of America) for 5 consecutive days at 8 weeks of age. Mice were phenotyped 2 weeks later. Successful and specific activation of the Cre recombinase has been verified by measuring recombination efficacy in *Cdh5-Cre^ERT2;Rosa26^mTmG* mice (S2A Fig). Importantly, Dhh endothelial knockdown does not impact CNS angiogenesis (S2 Data) nor angioarchitecture (S2D Fig). The *Cdh5-Cre^ERT2* mice and C57BL/6 mice were purchased from Jackson Laboratories (Bar Harbor, Maine, USA).

## Ethical statement

Human samples: The NeuroCEB bio bank and the INSERM U1034 certify that all human sections utilized for this study are ethically obtained with documented, legal permission for research use (authorization number #AC-2018-3290 obtained from the Ministry of Higher Education and Research) and in the respect of the written given consent from the source person in accordance with applicable laws and the World Medical Association (WMA) Helsinki declaration of 2013.

Animal experiments were performed in accordance with the guidelines from Directive 2010/63/EU of the European Parliament on the protection of animals used for scientific purposes and approved by the local Animal Care and Use Committee of the Bordeaux University CEEA50 (IACUC protocol #16901).

## Neurovascular fraction enrichment from mouse CNS

Mouse was humanely killed by cervical dislocation, and its head was cut and rinsed with 70% ethanol. Brain and spinal cord were then harvested, and cerebellum, olfactory bulb, and white matter were removed from the brain with sterile forceps. Additionally, meninges were eliminated by rolling a sterile cotton swab at the surface of the cortex. The cortex and spinal cord were then transferred in a potter containing 2 mL of buffer A (HBSS 1X w/o phenol red (Gibco, Waltham, Massachusetts, USA), 10-mM HEPES (Gibco), and 0.1% bovine serum albumin (BSA) (Sigma Aldrich), and the CNS tissue was pounded to obtain an homogenate, which was collected in a 15-mL tube. The potter was rinsed with 1 mL of buffer A, which was added to the 2-mL homogenate. Cold 30% dextran solution was then added to the tube (V:V) to obtain a 15% dextran working solution centrifuged for 25 minutes at 3,000 *g*, 4˚C without brakes. After centrifugation, the pellet (neurovascular components and red cells) was collected, and the supernatant (dextran solution and neural components) was centrifuged again to get the residual vessels. Neurovascular components were then pooled and resuspended in 4 mL of buffer B (HBSS 1X $Ca^{2+}$/ $Mg^{2+}$ free with phenol red (Gibco), 10-mM HEPES (Gibco), and 0.1% BSA (Sigma Aldrich)).

## Neurovascular fraction enrichment for RT-PCR, western blots, or immunohistochemistry

After centrifugation of the cell suspension, the pellet was washed 3 times with the buffer B and filtered through a 100-μm nylon mesh (Millipore Corporation, Burlington, Massachussetts, USA). The nylon mesh was washed with 7 mL of buffer B to collect the retained enriched

neurovascular fractions. The suspension was then centrifuged for 10 minutes at 1,000 $g$, and the pellet suspended in 300 μL of radioimmunoprecipitation assay (RIPA) lysis buffer for western blot analysis or 1,000 μL of Tri-Reagent (MRC, Cincinnati, Ohio, USA) for quantitative reverse transcription polymerase chain reaction (qRT-PCR) analysis. For immunohistochemistry, the pellet was suspended in 3 mL of a solution of matrigel (Corning, Steuben, New York, USA)-Dulbecco's Modified Eagle Medium (DMEM) 1 g/L glucose, $Mg^+$, $Ca^{2+}$ (Gibco) 1:80, distributed on a labtek (Starstedt, Nümbrecht, Germany) (1 mouse brain is needed to seed 1 labtek) and incubated for 30 minutes at 37°C. Finally, the enriched neurovascular fraction embedded in the matrigel (Corning) solution was fixed with 10% formalin for 10 minutes.

### Primary culture of mouse CNS micro vascular endothelial cells (CNS MECs)

After centrifugation of the cell suspension, the pellet was washed 3 times with the buffer B and transferred in an enzyme solution (2 mg/mL Collagenase/Dispase (Roche, Bale, Switzerland), 0.147 μg/mL TLCK (Lonza, Bäle, Switzerland), and 10 μg/mL DNAse 1 (Roche)), prewarmed at 37°C, before being placed on a shaking table at maximum speed agitation at 37°C. After 30 minutes, the digestion was stopped by adding 10 mL of buffer B, the cell suspension centrifuged, and the digested neurovascular pellet washed 3 times with 3 mL of buffer B. After the 3 washing steps, the digested neurovascular pellet was resuspended in 1 mL of Mouse Brain Endothelial Cell Culture Medium (DMEM 1 g/L glucose, $Mg^+$, $Ca^{2+}$ (Gibco), fetal bovine serum (FBS) 20% (Gibco), sodium pyruvate 2% (Gibco), nonessential amino acids 2% (Lonza), FGF 1 ng/mL (PeproTech, Rocky Hill, New Jersey, USA), and gentamycin 10 mg/mL (Gibco)), and plated on a labtek (Starstedt) (1 mouse brain is needed to seed 1 labtek) previously coated with 2% matrigel (Corning) diluted in DMEM 1 g/L glucose, $Mg^+$, $Ca^{2+}$ (Gibco).

### Cell culture

HBMECs (Alphabioregen–CliniSciences, Nanterre, France) were cultured in endothelial basal medium-2 (EBM-2) supplemented with EGM-2 BulletKits (Lonza). Cells from passage 3 to passage 6 were used. Before any treatment, cells were serum starved in 0.5% FBS EGM-2 medium for 24 hours. NHAs (Lonza) were cultured in astrocyte basal medium (ABM) supplemented with AGM BulletKits (Lonza). Cells from passage 2 to passage 4 were used. Before any treatment, cells were serum starved in DMEM 1 g/L glucose, $Mg^+$, $Ca^{2+}$ (Gibco) without serum for 24 hours.

### Cytokines/growth factors/chemicals

Human IL-1β was purchased from PeproTech (Rocky Hills, New Jersey, USA), and Human and mouse VEGF-165 (VEGFA) were purchased from CliniSciences (Nanterre, France). Based on previous studies, Human IL-1β and Human VEGF-165 were routinely used at 10 ng/mL [19,49]. Mouse VEGF-165 was used at a concentration of 20 ng/μL. The inhibitor of FOXO1 total (AS1842856) was purchased from Merck (Kenilworth, New Jersey, USA) and was used at 100 nM [50]. D-Mannitol was purchased from Sigma Aldrich (St. Louis, Missouri, USA) and was used at 100 mM [51].

### Antibodies

Anti-CLDN4 (mouse), anti-CLDN5 (mouse (tissues) and rabbit (cell culture)), anti-human ZO1 (rabbit), and anti-GFAP (rat) were from Invitrogen (Carlsbad, California, USA). Anti-CDH5 (goat) was from R&D systems (Minneapolis, Minnesota, USA). Anti-human CDH5 (mouse) and anti-human DHH (H-85) (rabbit) were from Santa Cruz Biotech (Santa Cruz,

California, USA). Anti-FGB (rabbit) and anti-human PECAM1 (mouse) were from Dako (Carpinteria, California, USA). Anti-ALB (sheep) and anti-MBP (rat) were from Abcam (Cambridge, Massachusetts, USA). Anti-CD45 (rat) was from eBioscience (San Diego, California, USA). Anti-RNA binding fox-1 homolog 3 also known as neuronal nuclei antigen (NEUN) (rabbit) and anti-AQP4 (rabbit) were from Millipore (Billerica, Massachusetts, USA). Anti-LAM (rabbit) was from Sigma Aldrich (St. Louis, Missouri, USA). Anti-ZO1 (rabbit) was from Life Technologies (Grand Island, New York, USA). Anti-FOXO1 (rabbit), anti-p-FOXO1 (rabbit), and anti-β-ACTIN (rabbit) were from cell signaling (Danvers, Massachusetts, USA).

## Quantitative RT-PCR

RNA was isolated using Tri Reagent (Molecular Research Center) as instructed by the manufacturer, from $3 \times 10^5$ cells or from isolated mouse-enriched neurovascular fractions. For qRT-PCR analyses, total RNA was reverse transcribed with Moloney Murine Leukemia Virus (M-MLV) reverse transcriptase (Promega, Madison, Wisconsin, USA), and amplification was performed on a DNA Engine Opticon2 (MJ Research, St. Bruno, Canada) using B-R SYBER Green SuperMix (Quanta Biosciences, Beverly, Massachusetts, USA). Primer sequences are reported in Table 1.

The relative expression of each mRNA was calculated by the comparative threshold cycle method and normalized to *β-actin* mRNA expression.

## Western blots

Protein expression was evaluated by SDS-PAGE. Protein loading quantity was controlled using the rabbit monoclonal anti-β-actin antibody (cell signaling). Secondary antibodies were from Invitrogen. The signal was then revealed by using an Odyssey Infrared imager (LI-COR, Lincoln, Nebraska, USA). For quantification, the mean pixel density of each band was measured using Image J software (NIH, Bethesda, Maryland, USA), and data were standardized to β-actin, and fold change versus control calculated.

## Stereotactic injection

Ten-week-old C57BL/6 mice (4 mice per condition) were anaesthetized using isoflurane (3% induction and 1% maintenance) (Virbac Schweiz, Glattbrugg, Germany) and placed into a stereotactic frame (Stoelting Co., Illinois, USA). To prevent eye dryness, an ophthalmic ointment was applied at the ocular surface to maintain eye hydration during the time of surgery. The skull was shaved, and the skin incised on 1 cm to expose the skullcap. Then, a hole was drilled into the skull, using a pneumatic station S001+TD783 Bien Air, until reaching the dura mater. A total of 3 μl of murine VEGFA (20 ng/μL), 3 μL of healthy mouse plasma, or 3 μL of vehicle control (PBS) were then delivered at 0.01 μl/s into the frontal cortex at coordinates of 1 μm posterior to bregma, 2 μm left of the midline, and 1.5 μm below the surface of the cortex [36].

Mice received a subcutaneous injection of buprenorphine (0.05 mg/kg) (Ceva santé animale, Libourne, France) 30 minutes before surgery and again 8 hours post-surgery to assure a constant analgesia during the procedure and postoperatively. Mice were humanely killed by pentobarbital (Richter Pharma, Wels, Austria) overdose at 24 hours post injection (dpi). For histological assessment, the brain of each animal was harvested.

## Experimental autoimmune encephalomyelitis (EAE)

Ten-week-old female mice were immunized by subcutaneous injection of 300-μg myelin oligodendrocyte glycoprotein-35-55 (MOG$_{35-55}$) (Hooke Laboratories, Lawrence, Massachusetts,

**Table 1. List of primers used for RT qPCR.**

| mDhh | F | 5′ -CTTGGACATCACCACGTCTG- 3′ |
|---|---|---|
| | R | 5′ -ATGTAGTTCCCTCAGCCCCT- 3′ |
| mIcam1 | F | 5′ -TGGCCTGGGGGATGCACACT- 3′ |
| | R | 5′-CCACCGGGCTGTAGGTGGGT-3′ |
| mVcam1 | F | 5′ -CGTACACCATCCGCCAGGCA- 3′ |
| | R | 5′ -TAGAGTGCAAGGAGTTCGGGCG- 3′ |
| mCldn5 | F | 5′ -GCAAGGTGTATGAATCTGTGCT- 3′ |
| | R | 5′ -GTCAAGGTAACAAAGAGTGCCA- 3′ |
| mZo1 | F | 5′ -GCTAAGAGCACAGCAATGGA- 3′ |
| | R | 5′ -GCATGTTCAACGTTATCCAT- 3′ |
| mβ-actin | F | 5′ -GAAGCTGTGCTATGTTGCTCTA- 3′ |
| | R | 5′—GGAGGAAGAGGATGCGGCA- 3′ |
| hGfap | F | 5′ -TGGGTCAAAGGAAACCGGAA- 3′ |
| | R | 5′ -GAAAGTCCCAAGCCATCAGC- 3′ |
| hDhh | F | 5′ -AACAGCTTACTTCCGGCTCC- 3′ |
| | R | 5′ -CGACTCTTGTGGGCTCTGTT- 3′ |
| hIcam1 | F | 5′ -ACGCCGGAGGACAGGGCATT- 3′ |
| | R | 5′ -GGGGCTATGTCTCCCCCACCA- 3′ |
| hVcam1 | F | 5′ -GGCCCAGTTGAAGGATGCGGG- 3′ |
| | R | 5′ -AGAGCACGAGAAGCTCAGGAGAA- 3′ |
| hAldh1L1 | F | 5′ -CCAGGGTTCTTCTTTGAGCCA- 3′ |
| | R | 5′ -CACCAGAAGCCAGGCCAAAT- 3′ |
| hVim | F | 5′ -CAGTTTTTCAGGAGCGCAAGA- 3′ |
| | R | 5′ -CAAGTTGGTTGGATACTTGCTGG- 3′ |
| hCldn4 | F | 5′ -GACACTAATGAGCCTGGGAGG- 3′ |
| | R | 5′ -GTGCACAGGTCCCATTTATTGTAG- 3′ |
| hβ-actin | F | 5′ -GCTGTGCTACGTCGCCCTG- 3′ |
| | R | 5′ -GGAGGAGCTGGAAGCAGCC- 3′ |

F, forward; R, reverse; RT qPCR, quantitative reverse transcription polymer chain reaction.

*β-actin* was used as the household gene.

USA) in 200-μl Freund's Adjuvant containing 300-μg/mL mycobacterium tuberculosis H37Ra (Hooke Laboratories) in the dorsum. Mice were administered with 500-ng pertussis toxin (PTX) intraperitoneously on day of sensitization and 1 day later (Hooke Laboratories). The emulsion provides antigen, which initiates expansion and differentiation of MOG-specific auto-immune T cells. PTX enhances EAE development by providing additional adjuvant. EAE will develop in mice 7 to 14 days after immunization (Day 0): Animals that develop EAE will become paralyzed. Disease was scored (0, no symptoms; 1, floppy tail; 2, hind limb weakness (paraparesis); 3, hind limb paralysis (paraplegia); 4, fore and hind limb paralysis; 5, death)) [31] from day 7 post immunization until day 32 post immunization. At Day 32, all the animals were euthanized by pentobarbital (Richter Pharma) overdose. For histological assessment, cervical, lumbar, and dorsal sections of each animal spinal cord, as well as the spleen, were harvested.

## Immunohistochemistry

Prior to tissue collection and staining, mice were transcardially perfused with PBS (10 mL) followed by 10% Formalin (10 mL) to remove intravascular plasma proteins. Brain and spinal cord samples were either fixed in 10% formalin for 3 hours, incubated in 30% sucrose

overnight, OCT embedded and cut into 9-μm thick sections, or directly OCT embedded and cut into 9 μm thick sections. Cultured cells were fixed with 10% formalin for 10 minutes. Human frozen sections were used directly without any prior treatment. Concerning the fixed sections, for CLDN4, prior to blocking, sections were soaked in Citrate (pH 7.5; 100˚C). For CLDN5, prior to blocking, sections were soaked in EDTA (pH 6.0; 100˚C). For CD45, sections were treated with 0.5 mg/mL protease XIV (Sigma Aldrich) at 37˚C for 5 minutes. Primary antibodies were used at 1:100 except CLDN4 (1:50), FGB (1:1,000), and ALB (1:1,000). Samples were examined using a Zeiss Microsystems confocal microscope (Oberkochen, Germany), and stacks were collected with $z$ of 1 μm.

For immunofluorescence analyses, primary antibodies were resolved with Alexa Fluor–conjugated secondary polyclonal antibodies (Invitrogen), and nuclei were counterstained with DAPI (1:5000) (Invitrogen). For all immunofluorescence analyses, negative controls using secondary antibodies only were done to check for antibody specificity.

## Morphometric analysis

Morphometric analyses were carried out using NIH ImageJ software (NIH).

BBB permeability was evaluated by measuring tight junction integrity and plasmatic protein extravasation. Brain and spinal cord sections were immunostained for the expression of CLDN5/CDH5 and FGB/IgG/ALB, respectively. For each brain or spinal cord section, CLDN5+, CDH5+, FGB+, IgG+, and ALB+ areas were quantified in 20 pictures taken at the margins of the lesion area under 40× magnification. One section was quantified per spinal cord (3 different zones are displayed on the same section: 1 cervical, 1 lumbar, and 1 dorsal to get a global vision of the lesion) (per mouse).

Leukocyte densities were evaluated in sections stained for the expression of CD45 leukocyte population. For each brain or spinal cord section, CD45+ leukocytes were counted in 20 pictures randomly taken under 40× magnification. One section was quantified per spinal cord (3 different zones are displayed on the same section: 1 cervical, 1 lumbar, and 1 dorsal to get a global vision of the inflammatory lesion) (per mouse).

Demyelination was evaluated in spinal cord sections stained for the expression of MBP. For each spinal cord section, MBP+ area was quantified in 10 pictures taken in and around inflammatory lesion sites under 20× magnification. One section was quantified per spinal cord (3 different zones are displayed on the same section: 1 cervical, 1 lumbar, and 1 dorsal to get a global vision of the lesion) (per mouse).

Plasmatic protein and leukocyte infiltrate distribution at the neurovascular unit were evaluated in brain or spinal cord sections (1) triple stained for PECAM1 or CDH5 (markers of the BBB), IgG (plasmatic proteins), and AQP4 or GFAP (markers of the Glia Limitans); or (2) double stained for IgG (plasmatic proteins), FITC Dextran (exogenous tracer) or CD45 (leukocyte infiltrate), and LAM (marker of basement membranes). For each section, the distribution (between the lumen, the PVS, and the parenchyma) of IgG, 70 kDa FITC Dextran or leukocyte infiltrate was quantified for 5 to 6 neurovascular units randomly taken under 60× magnification, each 1 from a different animal. We used negative working images highlighting the endothelial BBB and astrocyte Glia Limitans and outlined them by using dotted lines. Dotted lines were then transferred to the plasmatic protein or leukocyte infiltration images so that we could quantify their distribution within the 3 compartments (lumen, PVS, and parenchyma).

## Statistical analyses

Results are reported as mean ± SEM. Comparisons between groups were analyzed for significance with the nonparametric Mann–Whitney U test, the nonparametric Kruskal–Wallis test

followed by the Dunn multiple comparison test when we have more than 2 groups, the chi-squared test for the distribution of plasmatic proteins and inflammatory cells in the neurovascular unit, or a nonlinear regression test (Boltzmann sigmoidal) for the EAE scoring analysis using GraphPad Prism v8.0.2 (GraphPad, San Diego, California, USA). Differences between groups were considered significant when $P \leq 0.05$ ($^*P \leq 0.05$, $^{**}P \leq 0.01$, $^{***}P \leq 0.001$).

## Supporting information

**S1 Text.** Supporting information file containing the S1 Data and S2 Data legends and DOI links (A), the Supporting Methods (B), and the associated References (C).
(DOCX)

**S1 Raw Images.** Supporting information file containing the original, uncropped, and minimally adjusted images supporting all blot and gel results reported in Fig 2 panel G, Fig 4 panel F, and Fig 7 panel I as well as S6 Fig panel A and B.
(PDF)

**S1 Fig. (Related to Fig 1 and Fig 2). BMEC purity analysis shows limited contamination by other neurovascular components (A)** Primary BMECs from Dhh^ECKO and control mice were isolated and cultured on Lab-Tek. ZO1 (in green) and SMA, NG2, IBA1, or GFAP (in red) localizations were evaluated by immunofluorescent staining of a confluent cell monolayer. Nuclei were stained with DAPI (in blue). The experiment was repeated 3×. **Mouse brain in situ hybridization analysis highlights *Dhh* expression at the BBB. (B)** C57BL/6 cortical cross section hybridized with the *Dhh* RNA probe show *Dhh* expression in blood vessels. A control section hybridized with the *Dhh* antisense RNA probe show the absence of hybridization signals. **MECs isolated from the spinal cord of mice induced with EAE MOG$_{35-55}$ are highly contaminated by leukocytes (C)** Spinal cord MECs were isolated from 12-week-old C57Bl/6 mice at day 13 post induction with EAE MOG$_{35-55}$ or placebo and *Pecam1*, *Cd45*, *Sma*, *Ng2*, *and Gfap* expressions were quantified by qRT-PCR (cycle threshold mean values) before and after the cell suspension were depleted in CD45$^+$ leukocytes. *β-actin* is used as a reference. **A MACS CD45+ cell depletion step is sufficient to eliminate leukocyte contamination in MECs isolated from the spinal cord of mice induced with EAE MOG$_{35-55}$ (D)** Representative graphs of flow cytometry analysis performed on C57BL/6 mouse spinal cord MECs harvested at day 13 post EAE MOG$_{35-55}$ induction. Analysis showed no CD45+ cell population in samples depleted in leukocytes using the MACS CD45+ cell isolation kit (Miltenyi Biotec) (EAE MOG$_{35-55}$ spinal cord MECs $n = 3$; EAE MOG$_{35-55}$ spinal cord MECs + MACS CD45+ cell isolation $n = 3$). **Dhh endothelial knockdown does not impact CNS MEC viability in culture (E-F). (E)** Primary CNS MECs from Dhh^ECKO and control mice were isolated and cultured on Lab-Tek and immunostained with Propidium Iodide (in red) and Hoechst 33342 to label the nuclei (in blue). **(F)** Dhh^ECKO versus control primary CNS MEC viability was evaluated by quantifying the number of nuclei that incorporated Propidium Iodide. NS, Mann–Whitney U test. The underlying data for S1 Fig can be found in S2 Data (https://doi.org/10.6084/m9.figshare.12625085.v7).
(TIF)

**S2 Fig. (Related to Fig 3, Fig 4, Fig 6 and Fig 7).** C*adherin5Cre^ERT2* recombinase activation in blood vessels is successful and specific **(A)** Brain and spinal cord sections were harvested from *Cadherin5Cre^ERT2*,*Rosa26^mTmG* mice and littermate controls and immunostained with anti-GFP (in green) and anti-PECAM1 (in red) antibodies. **Dhh endothelial knockdown does not impact CNS angiogenesis (B)** Spinal cord sections were harvested from Dhh^ECKO mice and littermate controls and immunostained with an anti-IB4 (in green) antibody. IB4 positive

area was quantified (Dhh$^{ECKO}$ $n$ = 7, control $n$ = 6). **(C)** Cortical sections were harvested from Dhh$^{ECKO}$ mice and littermate controls and immunostained with an anti-IB4 (in green) antibody. IB4 positive area was quantified (Dhh$^{ECKO}$ $n$ = 6, WT $n$ = 6). **Dhh endothelial knockdown does not impact brain angioarchitecture (D)** The vascular network in the brain of Dhh$^{ECKO}$ mice and control littermates was imaged by microcomputed tomography (micro-CT). NS, Mann–Whitney U test. The underlying data for S2 Fig can be found in S1 Data (individual numerical data (excel file)) and S2 Data (statistical analysis (Prism file)) (https://doi.org/10.6084/m9.figshare.12625034.v6; https://doi.org/10.6084/m9.figshare.12625085.v7). (TIF)

**S3 Fig. Endothelial-specific Dhh inactivation induces BBB permeability in vivo. (A)** Spinal cord sections were harvested from Dhh$^{ECKO}$ mice and littermate controls and immunostained with anti-IB4 (in green) and anti-CLDN5 (in red) antibodies. Representative IB4/CLDN5 staining was shown. **(B–C)** Spinal cord sections were harvested from Dhh$^{ECKO}$ mice and littermate controls and immunostained with anti-CDH5 or anti-CLDN5 (in green), and anti-FGB or anti-ALB (in red) antibodies. Representative **(B)** CDH5/FGB and **(C)** CLDN5/ALB staining were shown. (TIF)

**S4 Fig. (Related to Fig 4). Dhh$^{ECKO}$-induced BBB breakdown is sufficient to induce a secondary CNS protective barrier at the Glia Limitans. (A)** Brain sections were harvested from Dhh$^{ECKO}$ mice and littermate controls injected with 70 kDa FITC Dextran and **(A)** immunostained with an anti-LAM (in red) antibody (nuclei were stained with DAPI (in blue)). Representative LAM/FITC Dextran staining was shown. **(B)** Negative working images of LAM channels were used to highlight the endothelial (EBM) and astrocyte (ABM) basement membranes, using orange dotted lines. The outlines were then transferred to the FITC Dextran images to discriminate the distribution of FITC Dextran between the lumen, PVS, and parenchyma. **(C–D)** The distribution of FITC Dextran within the lumen, PVS, and parenchyma was quantified. (Dhh$^{ECKO}$ $n$ = 6, control $n$ = 6) $P$ < 0.0001, 95% CI: 0.3518 to 0.5734, chi-squared test. **(E)** Negative working images of AQP4/PECAM1 channels were used to highlight the endothelial (EBM) and astrocyte (ABM) basement membranes, using orange dotted lines. The outlines were then transferred to the IgG images to discriminate the distribution of IgG between the lumen, PVS, and parenchyma. The underlying data for S4 Fig can be found in S2 Data (https://doi.org/10.6084/m9.figshare.12625085.v7). (TIF)

**S5 Fig. (Related to Fig 4). Dhh$^{ECKO}$-induced BBB breakdown is sufficient to induce a secondary CNS protective barrier at the Glia Limitans. (A)** Spinal cord sections were harvested from Dhh$^{ECKO}$ mice and littermate controls and immunostained with anti-LAM (in green) and anti-IgG (in red) antibodies (nuclei were stained with DAPI (in blue)). Representative LAM/IgG staining was shown. **(B)** Negative working images of LAM channels were used to highlight the endothelial (EBM) and astrocyte (ABM) basement membranes, using orange dotted lines. The outlines were then transferred to the IgG images to discriminate the distribution of IgG between the lumen, PVS, and parenchyma. (TIF)

**S6 Fig. (Related to Fig 4). Small intestine samples are used as a positive control for the quantification of CLDN4 expression by western blot. (A)** Representative blots of CLDN4 expression level on control mouse neurovascular unit lysates and mouse small intestine lysates were shown. **There is astrocyte reactivity but no astrocytic CLDN4 up-regulation at the capillary level in Dhh$^{ECKO}$ mouse CNS. (B)** Representative blots of GFAP and CLDN4

expression level on Dhh^ECKO and control lysates were shown. Lysates were obtained with neurovascular units, which are 20 μm and larger. **(C)** GFAP expression level was quantified by western blot on Dhh^ECKO and control lysates obtained with neurovascular units, which are 20 μm and larger. **(D)** CLDN4 expression level was quantified by western blot on Dhh^ECKO and control lysates obtained with neurovascular units, which are 20 μm and larger. (Dhh^ECKO $n = 6$, WT $n = 6$). *$P \leq 0.05$, Mann–Whitney U test. The underlying data for S6 Fig can be found in S2 Data (https://doi.org/10.6084/m9.figshare.12625085.v7).
(TIF)

**S7 Fig. (Related to Fig 5 and Fig 7). Both VEGFA and Mannitol induce HBMEC permeability in vitro. (A)** Cultured HBMECs were treated with PBS, VEGFA, or Mannitol for 6 h, and HBMEC monolayer permeability to 70 kDa FITC Dextran was quantified. **Mice with endothelial Dhh knockdown display a reinforced barrier at the Glia Limitans restraining access to the parenchyma to inflammation in a model of multiple sclerosis: (B)** Negative working images of GFAP/CDH5 channels were used to highlight the endothelial (EBM) and astrocyte (ABM) basement membranes, using orange dotted lines. The outlines were then transferred to the FGB images to discriminate the distribution of FGB between the lumen, PVS, and parenchyma. **(C)** Negative working images of the LAM channel were used to highlight the endothelial (EBM) and astrocyte (ABM) basement membranes, using blue dotted lines. The outlines were then transferred to the CD45 images to discriminate the distribution of leukocytes between the lumen, PVS, and parenchyma. *$P \leq 0.05$, ****$P \leq 0.0001$ Kruskal–Wallis test. The underlying data for S7 Fig can be found in S2 Data (https://doi.org/10.6084/m9.figshare.12625085.v7).
(TIF)

## Acknowledgments

We thank Dr. Mary P. Heyer for her proofreading and correction of the manuscript. We thank Annabel Reynaud, Sylvain Grolleau, and Maxime David for their technical help. We thank Christelle Boullé for administrative assistance.

## Author Contributions

**Conceptualization:** Candice Chapouly.

**Data curation:** Pierre Mora, Pierre-Louis Hollier, Sarah Guimbal, Alice Abelanet, Aïssata Diop, Lauriane Cornuault, Marie-Ange Renault, Candice Chapouly.

**Formal analysis:** Pierre-Louis Hollier, Sarah Guimbal, Aïssata Diop, Lauriane Cornuault, Candice Chapouly.

**Funding acquisition:** Candice Chapouly.

**Investigation:** Candice Chapouly.

**Methodology:** Sam Horng, Candice Chapouly.

**Resources:** Candice Chapouly.

**Supervision:** Candice Chapouly.

**Validation:** Pierre Mora, Pierre-Louis Hollier, Sarah Guimbal, Alice Abelanet, Aïssata Diop, Lauriane Cornuault, Marie-Ange Renault, Candice Chapouly.

**Visualization:** Alice Abelanet.

**Writing – original draft:** Candice Chapouly.

**Writing – review & editing:** Thierry Couffinhal, Sam Horng, Alain-Pierre Gadeau, Marie-Ange Renault, Candice Chapouly.

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
