## [Editor Report · Decision Letter 0]

1 Apr 2020

Dear Dr Chapouly, 

Thank you for submitting your manuscript entitled "Blood Brain Barrier genetic disruption leads to protective barrier formation at the Glia Limitans" for consideration as a Research Article by PLOS Biology.

Your manuscript has now been evaluated by the PLOS Biology editorial staff [as well as by an academic editor with relevant expertise] and I am writing to let you know that we would like to send your submission out for external peer review.

Please re-submit your manuscript within two working days, i.e. by Apr 03 2020 11:59PM.

Kind regards,

Di Jiang,

Associate Editor

PLOS Biology

---

## [Decision Letter · Decision Letter 1]

30 Apr 2020

Dear Dr Chapouly,

Thank you very much for submitting your manuscript "Blood Brain Barrier genetic disruption leads to protective barrier formation at the Glia Limitans" for consideration as a Research Article at PLOS Biology. Your manuscript has been evaluated by the PLOS Biology editors, an Academic Editor with relevant expertise, and by three independent reviewers.

In light of the reviews (below), we will welcome re-submission of a much-revised version that addresses the reviewers' concerns including, particularly, the important controls requested by reviewers 1 and 3, which should be included in the revision. We cannot make any decision about publication until we have seen the revised manuscript and your response to the reviewers' comments. Your revised manuscript is also likely to be sent for further evaluation by the reviewers.

We expect to receive your revised manuscript within 2 months. Please note given we are in the midst of COVID-19, we are flexible regarding turnaround time for revision.

**IMPORTANT - SUBMITTING YOUR REVISION**

*Re-submission Checklist*

*Published Peer Review*

*PLOS Data Policy*

*Blot and Gel Data Policy*

Sincerely,

Di Jiang, PhD

Associate Editor

PLOS Biology

REVIEWS:

Reviewer #1: 

Previous work from other labs have highlighted the concept of the double blood brain barrier or more accurately, the double-basement membrane structure of cerebral blood vessels (endothelial basement membrane and parenchymal basement membrane made by the glia limitans). Results suggested that penetration of the endothelial and parenchymal barriers are independent steps involving distinct molecular mechanisms.

In this study, the authors investigated this concept in a mouse model with conditional inactivation of Desert Hedghog (Dhh) in basal condition and after immunization by MOG, as a model of experimental encephalomyelitis. Although the concept is interesting, there are several major issues which limit the significance of this paper. 

Figure 1:

On panel A, positive and negative controls are missing to assess the purity of the preparation and confirm that these are endothelial cells. The authors jump throughout the paper from murine to human cells without any justification. Please explain. 

In situ hybridization on murine brain sections is needed to confirm production of Dhh by endothelial cells. 

On panels F-H, it is unclear whether expression has been analyzed in isolated microvessels or in cultured brain endothelial cells. Authors analyzed expression of Dhh 16 days after MOG immunization, which is quite late. Please explain why this late time point has been selected.

Figure 2 and 3

The conclusion that expression of VE-cadherin and claudin 5 is decreased in Dhh KO mice is unfortunately not convincing. 

Figure 2: There is a discrepancy between immunostaining (panels D-F) and qRT-PCR or Western blot data, regarding the expression of VE-cadherin and claudin 5. IF pictures show an increased expression of VE-cadherin and claudin 5 in mutant cells whereas qRT-PCR and WB quantifications show the opposite. Moreover, what is the genotype of control mice? Were these mice treated with tamoxifen? Also, same comment as above regarding brain endothelial cells vs isolated microvessels. 

Figure 3: On panels A -C, what is missing is a vascular marker to quantify claudin5 and VE-cadherin area over the vessel area, to normalize for the number and length of vessels. 

Methodology to analyze the BBB integrity in Dhh KO mice is not appropriate. At least mice must be transcardially perfused to remove intravascular plasma proteins which can passively diffuse in the vessel wall even after fixation (see Mestre et al, Nature Communications 2018). Moreover, because quantification of endogenous fibrinogen or albumin on sections can be challenging and is subjective, analysis of exogenous tracers such as cadaverine of Dextran is requested. 

Figure 4

The conclusion that plasma leakage in Dhh KO mice is restricted by the glia limitans to the perivascular space is unfortunately not convincing. What is missing is a labelling with a pan laminin antibody to identify the endothelial and parenchymal basement membranes as described in Song et al, Cell reports 2015. Moreover, aquaporin 4 would have been a better marker of astrocytic endfeet. Indeed, as shown in panel D, GFAP labelling of astrocytic endfeet is inconstant. Also, podocalyxin, which is expressed on the luminal membrane of endothelial cells, cannot delineate the abluminal wall of the vessel. 

The conclusion that "BBB breakdown is sufficient to induce a secondary CNS protective barrier at the Glia Limitans » is not supported by the data in the absence of the analysis of other mouse models with BBB leakage. This might depend on the cause of BBB leakage. 

Another problem is that authors did not differentiate vessel types in their analysis. The concept of double-basement membrane structure of cerebral blood vessels makes sense in arteries/ arterioles or veins/venules but not in capillaries where the parenchymal and vascular basement membranes have merged. 

Figure 5

Authors used VEGF or mannitol to simulate BBB leakage in vitro in cultured cells but there is no attempt to provide evidence that VEGF or mannitol did really something on their cells. 

Again, there is a discrepancy between GFAP staining shown in panels E-H and GFAP quantification. 

In the in vivo experiment the "hole drilled into the cerebral cortex", as mentioned in the material and methods, is likely to have induced unspecific astrocytic changes 

Figure 6

There seems to be a discrepancy between the alleged beneficial effect of inactivating Dhh on demyelination and the comparable lethality, unless the effect is marginal. Please explain. 

Figure 7

Panel E shows a diffuse astrocytic reaction in Dhh KO mice whereas fibrinogen staining seems to be restricted around what seems to be an arteriole. How do the authors reconcile this finding with their conclusion? 

Also, in panel H vessels from the WT and Dhh KO mice are clearly different, with no perivascular cuff in the WT and a perivascular cuff in the KO mouse. Quantification of the number of CD45+ and T cells inside or outside this cuff is valid, provided that comparable vessels (arteries or veins) are analyzed in each genotype. 

Others

In panel B (Fig 4) and in panels F-G, I-J (Fig 7) authors mention % values, but the confidence interval is not indicated. Also, it is very unclear how statistics have been performed. Comparison of % is usually analyzed by a Chi-square test but this is not mentioned in the Method section.

Reviewer #2: This paper by Hollier et al describes the functional emergence of a 2nd barrier formed by the glial limitans (astrocytes) following BBB breakdown in neuroinflammation. Molecularly, show that glial limitans 'tightening' to prevent parenchymal invasion of immune cells is a consequence of signals triggered by BBB breakdown (do this using a genetic model BBB breakdown, conditional deletion of Dhh and in MS animal model, also provide evidence in human MS tissue). This is very intriguing concept and builds off their 2017 JCI paper showing upregulation of astrocytic Cldn4 in MS animal models that helps to functional blocks parenchymal immune cell invasion that promotes disease progression. In general, I think this is a well-done study and of high interest to the fields of BBB mechanisms and neuroinflammatory disease. Below are my moderate concerns about some aspect of their analysis and a list of study limitations that should be addressed in the Discussion. 

Major: 

Line 115: Perivascular spaces are 'seen' around large penetrating arteriole and venules but I would check the literature carefully as I am not aware of the experimental evidence for PVS around capillary vasculature, which makes up the majority of vessels in the brain. At the capillary level, the astrocytes make direct and essentially continuous contact/coverage of abluminal surface. This becomes important later as the authors are proposing to analyze the PVS - laying out the literature of the existence of a space is important to support the validity of this analysis approach 

Ln 191-193, Fig. 2: These are qualitative statements that should be supported by quantitative analysis of junctional disorganization - this data is similar to what was described in their 2018 CR paper, I would be satisfied if they stated this is consistent with previously documented phenotype using siRNA for Dhh (junctional disorganization was quantified in this previous paper and looks similar) 

Fig. 2: This group showed something similar in the 2018 paper Circulation Research paper, that paper also documented altered organization of Cdh5 at the junctions (with siRNA) but not a decrease in expression (but could be for experimental reasons) - the reduction of Cdh5 in these mutants is unclear, nor is the model of how Dhh is regulating this. For example they show Cdh5 down in their cultures, the mutants but why? This isn't discussed, just the decrease in Cldn5. 

Fig. 3F: Please provide a low magnification image (can show current image as inset) to better demonstrate the scope of elevated ICAM expression. If the authors have this data, the ICAM with a blood vessel marker (PECAM) would help show this is vascular ICAM and how widespread it is in the spinal cord of the mutants. 

Fig. 3G: Based on the image, it seems like the increase in GFAP is limited to the white matter, did quantification include all areas of spinal cord (grey and white)? This would be important to state in the methods and state if the increase was or was not regionalized (could do that by segregating/presenting the data by region). 

Fig. 4G - There is insufficient detail in the methods as to how this was quantified. Recommend showing higher magnification images, also highlighting using dotted lines outlines where measurements were taken (possibly a separate diagram or image with this information). In it's current form, it's very difficult for me to see how accurate this type of analysis is for differentiated between fibrinogen in and outside the vessel. Also, the 24% in the control is a little concerning, the expectation would this is zero, again leading to some concerns about this method. A limitation of this assay is the that fibrinogen is very large, possible there is a size restriction feature of the glial limitans barrier - (70kda FITC dextran appeared to leak well away from the vessel in the 2018 paper) - two options, perform experiments with a smaller size dye to test this or discuss this as a limitation of their findings in the Discussion (possibly small size proteins could leak through the astrocytic barrier and still cause problems?)

Ln 313-315 Fig. 5K-O: Need to show that VEGFA injection into the cortex causes BBB breakdown - or cite papers that have used this method before and shown this method is effective

Fig. 7 - this data is confusing as presented w/o the non-EAE data - to understand the effects of EAE model on the two genotypes, non-EAE values for each of these parameters should be included on the histograms - in comparing data from Fig 3 and Fig 7, a lot of the same measures were done - was the analysis done the same and therefore could it be added here and stats run to make comparisons?

Ln 373: this is difficult to discern because Cdh5 expression is down, making it difficult to make out what is GFAP expression at glial limitans - remove statement or provide images/data that demonstrates this more clearly

Fig. 7F, G: Similar comment to this type of analysis in Fig. 4A/B, need much better description in methods for how this was done, a diagram or annotated imaging showing where analysis was performed - while it's very clear from the images that fibrinogen is 'trapped' it's not clear how the quantitative data was acquired

Ln 410: A model would be very helpful to bring all the observations of this paper together

Ln 429-431: I looked at Dhh and its expression in this dataset and it's actually very low vs other known endothelial genes - there could be MANY reasons for this, this was all from cortex, maybe there are regional differences, single cell can be tricky to pick up morphogens…I think the data in this manuscript is much more convincing of Dhh expression therefore I caution the authors against citing this as support for Dhh gene expression in brain endothelium

Ln 454-464 - One of the elements missing from this discussion is whether the increase in Cldn4 and astrocyte barrier induction is specific to the Dhh cKO model or is more broadly applicable to any mode of barrier opening - example in pericyte-deficient mice with BBB opening there is some problems with astrocyte-endfeet localization and the barrier opening is due to increased transcytosis no junctional degradation - bringing up these considerations is important 

Discussion: Another point that should be addressed in the discussion is whether this is a VEGFA driven process or broadly applicable to other BBB disrupters, example cytokines like IFNg - VEGFA but not mannitol was able to induce secretion of factors that signaled to astrocytes to increase Cldn4, what does this mean?

Discussion: Another limitation to this study that should be discussed is it's not clear if this is a general function of Dhh to regulate the barrier or region specific, all analysis was done in the spinal cord. Further, it's also important to consider if the upregulation of astrocytic Cldn4 is limited to neuroinflammatory disease. This and their previous paper have only looked at EAE in spinal cord. What is the evidence that this could be MS specific phenomenon vs generalizable reaction to BBB opening in other context of other actute of chronic disease (stroke, TBI, AD, ALS, etc). For example, can point to their own data that VEGFA injection into the cortex induced Cldn4 upregulation in response to BBB disruptor VEGFA. Or have other studies looked at this Cldn4 upregulation in response to BBB opening?

Minor:

Ln 109: Edit sentence for clarity; also I would argue that this vision of the NVU has been building for years so suggest removing the term 'recent' and replace with language that reflects a wealth of literature that builds this idea of a multi-cellular 'BBB'.

Ln 114: I would recommend adding a little more context for the reader - maybe a sentence saying this is a highly regulated process (solute transporters and receptor mediated trancytosis) and immune cells are actively prevented from cross by low levels immune receptors that normally permit immune trafficking.

Ln 119: consider a different phrase, perhaps "is more complex"

Ln 185: Please briefly describe how Dhh was deleted from ECs - are these ECs derived from the Cdh5-CreErtl Dhh1-fl/fl mice?

Ln 236: edit "BBB" to say brain endothelium since ICAM is expressed by the endothelial cells not the BBB per se.

Ln 271: "human disease" instead of "Human"

Ln 281: typo?

Ln 420: edit "proved" to say "showed"

Ln 420-424: break up in to two sentences

Reviewer #3: In the present study Chapouly and her team provide convincing evidence for an additional role of the glia limitans in regulating CNS immunity ,when BBB integrity is impaired. They have focused on the role of Dhh and show when deleted from the endothelium in mice, BBB tightness is impaired and at the same time tightness of the glia limitans is increased possibly by the induction of tight junctions in astrocytes. DhhECKO mice develop delayed EAE with CD45+ cells found to be trapped in perivascular spaces underscoring the notion of an increased barrier property of the glia limitans in these mice. 

The study combines analysis of human and mouse tissues and the finding of the cross-talk of an impaired BBB leading to improved barrier properties of the glia limitans to immune components is quite novel. 

The study as it stands has however a significant number of issues that need to be addressed. Overall the authors seem to tend towards conclusions that are not yet supported by the data as presented. Thus overall downtoning of the manuscript will be required. 

I am absolutely aware that in the COVID-19 pandemic a number of issues may not be solvable as they require additional experiments. If additional data cannot be provided the authors need to explain the reason and instead significantly downtone their conclusions. 

The authors base their study on the statement that "Presently, it is accepted that the BBB is the sole line of defense of the CNS, restricting access to the parenchyma to inflammatory infiltration notably in the context of chronic neuroinflammation." This is certainly not the case and the authors should include mentioning and discussion of the work of e.g. the Sorokin laboratory, which has provided mechanistic insight into the role of the glia limitans in CNS inflammation. Also Joan A Abbott has written several articles highlighting the role of the glia limitans as a brain barrier. In numerous reviews including those of Weller, Owens, Ransohoff and Engelhardt data on the role of the the glia limitans has been highlighted as barrier for the immune system. 

Throughout the manuscript the authors make repeatedly statements as the following that need to be corrected as suggested: 

* that the adherens junction Cdh5 and tight junction Cldn5 are down regulated

o downregulation of mRNA of junctional molecules does not allow to make any statement on the integrity or overall molecular composition of a cellular junction

* Isolation of neurovascular unit

o This unit of cellular and acellular components can hardly be isolated 

* Gliovascular unit

o what do the authors mean with this expression as compared to neurovascular unit?

o a network of astrocytes termed the Glia Limitans. 

o The glia limitans is composed of the parenchymal basement membrane and the astrocyte end-feet rather than a network of astrocytes. 

endothelial adherens and tight junction expression is maintained in DhhECKO

 this could only be stated when functional and ultrastructural studies confirm this

Comments to the data in the sequence of the manuscript: 

In general the authors refer to the gene names when staining or detecting proteins in IF stainings or WB -this is not correct and needs to be corrected. 

Figure 1 A shows lack of expression of Shh and Ihh in the healthy brain of mice - this is in apparent contrast to the observations by the Prat lab that has identified astrocytes as source of Shh maintining BBB stability, which is mentioned by the authors. The authors should compare the primer sets used and either repeat the assay or refer to the apparent difference. 

Figure 1 B does not show any overlap of VE-cadherin staining with the Dhh staining, nevertheless the authors conclude that Dhh is expressed in CNS endothelium. 

Figure 1F to J: Isolation of spinal cord microvessels from mice suffering from EAE will contain other contaminants as microvessels isolated from healthy controls. Lower levels of Dhh, Claudin-5 and ZO-1 from the vascular compartment accompanied by higher levels of ICAM-1, VCAM-1 could also be obtained by coisolation of a higher number of contaminations with astrocyte end-feet, perivascular inflammatory cells or their debris or regulated expression in pericytes and smooth muscle cells. In this context the author refer to their results in Fig 1, I-J showing lower levels of mRNA for claudin-5 and ZO1, as "down regulation of tight junctions"- this is inappropriate as tight junctions are complex cell-cell-contacts including more than describing the mRNA expression of their proteins. 

Figure 2A - The authors verify endothelial cell specific deletion of Dhh in vivo by determining mRNA levels for Dhh in endothelial cell cultures derived from these mice. This does of course not allow to verify the efficiency of Dhh depletion in vivo as it does now allow to determine if Dhh has been deleted with different efficiency from different parts of the CNS vasculature. It could thus be that the authors culture endothelial cells from different parts of the vascular tree as they also don't know if lack of Dhh has any effect in survival of the ECs during isolation or their growth in vitro. Furthermore, considering that Dhh is a morphogen it is mandatory to investigate how Cre-mediated deletion of Dhh affects brain angiogenesis. 

Figure 3: The authors refer to ICAM-1 and GFAF as markers analyzing the "activation status of both the BBB and Glia Limitans". As outlined several times using a reference molecule to describe an entire tissue or cell is not appropriate. Although ICAM-1 is an excellent marker for EC activation, in the context of the inflamed CNS tissue the authors have to consider that ICAM-1 is also upregulated on pericytes, smooth muscle cells and astrocytes. 

Figure 3A - why do the authors show two sets for VE-cadherin/Fibrinogen staining? It would be preferable to show another set e.g. for claudin-5 and albumin staining. 

Figure 4B: The authors have used podocalyxin and GFAP to define localization of fibrinogen in the lumen, perivascular space and within the CNS parenchyma. The perivascular space is bordered by basement membranes, which can be detected by pan-laminin stainings which would be preferable for accurate analysis of the localization of fibrinogen. The authors thus either perform additional stainings or significantly downtone this paragraph as GFAP as an intermediate filament, which is also not detectable in all astrocytes does not allow for accurate definition of the glia limitans. 

Claudin-4 is suggested to be expressed in GFAP + astrocytes- the staining for both markers looks to a large degree identical, which is confusing as GFAP is an intermediate filament whereas claudin-4 would be expected at the cell membranes and possibly focused to astrocyte end feet. Did the authors check for other claudins and could confirm specificity of this staining? 

Figure 4 E: WB signal for claudin-4 looks very weak. A positive control with claudin-4 transfectants would be helpful to show. 

Figure 5: The authors delivered VEGF-A or I guess what they mean plasma proteins into the left cerebral cortex of adult mice. They have used a very high volume of 3 μL which in addition was injected in only 30 seconds !!!! into the tissue - this will cause massive damage of the tissue, which needs to be considered when interpreting these data. 

Figure 6: The authors show delayed onset of EAE in the DhhEC KO mice. As mentioned above without prior analysis of the impact of endothelial cell specific deletion of Dhh in the entire vasculature of these mice, conclusions from these data are difficult to make. CNS angioarchitecture is the minimum that needs to be studied, but vascular deletion of Dhh could have effects beyond the CNS not considered by the authors but still impacting on EAE which is induced by peripheral activation of T cells that need to travel to the CNS. 

Last but not least the manuscript does need editing with the help of a native speaker.

---

## [Decision Letter · Decision Letter 2]

24 Sep 2020

Dear Dr Chapouly,

Thank you for submitting your revised Research Article entitled "Blood Brain Barrier genetic disruption leads to protective barrier formation at the Glia Limitans" for publication in PLOS Biology. I have now obtained advice from the original reviewers and have discussed their comments with the Academic Editor. You will note that reviewer 2, Julie Siegenthaler, has revealed her identity. 

Based on the reviews, we will probably accept this manuscript for publication, assuming that you will modify the manuscript to address the remaining points raised by the reviewers. Having discussed their specific comments with the Academic Editor, we think that toning down some of the conclusions is sufficient and no new experiments are required. 

Please also make sure to address the data and other policy-related requests noted at the end of this email.

We expect to receive your revised manuscript within two weeks. Your revisions should address the specific points made by each reviewer. In addition to the remaining revisions and before we will be able to formally accept your manuscript and consider it "in press", we also need to ensure that your article conforms to our guidelines. A member of our team will be in touch shortly with a set of requests. As we can't proceed until these requirements are met, your swift response will help prevent delays to publication.

- a cover letter that should detail your responses to any editorial requests, if applicable

*Copyediting*

*Published Peer Review History*

*Early Version*

Sincerely,

Gabriel Gasque, Ph.D.,

Senior Editor,

ggasque@plos.org,

PLOS Biology

ETHICS STATEMENT:

-- Please in your manuscript the ID number of the experimental protocol approved by your institute’s animal welfare officer and the Landesamt für Verbraucherschutz und Lebensmittelsicherheit (LAVES)

DATA POLICY:

Note that we do not require all raw data. Rather, we ask for all individual quantitative observations that underlie the data summarized in the figures and results of your paper. For an example see here: http://www.plosbiology.org/article/info%3Adoi%2F10.1371%2Fjournal.pbio.1001908#s5

These data can be made available in one of the following forms:

Regardless of the method selected, please ensure that you provide the individual numerical values that underlie the summary data displayed in the following figure panels: Figures 1C-J, 2A-C, 2H-L, 3C-F, 3I-K, 4CGH, 5A-D, 5IJNO, 6A-D, 6F, 7A-D, 7FHJ, S1F, S2BC, S4D, S6CD, and S7A.

Please also ensure that each figure legend in your manuscript include information on where the underlying data can be found and that your supplemental data file/s has/have a legend.

Reviewer remarks:

Reviewer #1: In their revised manuscript, Mora et al., have added new data and clarified some issues. However, important concerns remain to be resolved. 

In figure 2, the finding that CLDN5 is downregulated in MEC from Dhh KO mice is still not convincing. Contrary to what is mentioned in the rebuttal, expression of CLDN5 has not been analyzed by immunoblot and the panel F clearly shows an upregulation of CLDN5. 

The conclusion that plasma leakage in Dhh KO mice is restricted by the glia limitans to the PVS is still not convincing. 1) In figure 4, panel A, there is clearly an uptake of IgG in the astrocyte labeled by the anti-aquaporin 4 antibody. 2°) If the glia limitans restricts plasma leakage in DhhKO mice, then fibrinogen staining in Figure 3 panel shall be exclusively detected in the PVS around arteries or veins but not around capillaries, where there is a single composite basement membrane produced by endothelial cells, pericytes and astrocytic and no PVS. This does not seem to be the case in this figure 3, quality of which is quite poor.

On page 11, authors claim that Dhh inactivation at the BBB drives endothelial and astrocytic activation. Since they did not identify astrocytic activation in the gray matter (figure 3, panels H and K), this implies that there is no BBB leakage in the gray matter but this is not shown. Otherwise, this conclusion shall be toned down. 

In general the quality of immunofluorescence images is rather low and should be improved. 

Reviewer #2, Julie Siegenthaler: This is a revised version of Hollier et al probing the molecular basis of a 2nd barrier system formed by the glial endfeet in response to neuroinflammation. The authors have done an excellent job addressing my previous comments, including greatly improved or completely new quantitative analysis (GFAP in SC sections Fig. 3, IgG leakage in perivascular spaces Fig. 4/Supp Fig. 4, Fig. 7), also adding a helpful diagram to bring the data in non-disease vs EAE together for the reader. The authors also added new discussion points requested that are important to understand 1) is BBB breakdown a universal inducer of Cldn4 and 'strengthening' the glial limitans, 2) is the induction of Cldn4 specific to VEGFA-mediated disruption or more broadly applicable to other cytokines and 3) is this a MS/EAE phenomenon. 

I did note a typo in Figure 3J should be 'White' and the text in the summary figure is rather small, enlarging would help.

I read through the other reviewer comments and the authors' response and I want to commend the authors for their efforts in addressing what were VERY extensive comments, especially considering lab shut downs and general pandemic related upheaval. Great job!

Reviewer #3: The authors have thoroughly answered all queries. They should however add the source of the VE-cadherin-CreERT2 transgenic mouse in M&M.

---

## [Editor Report · Decision Letter 3]

22 Oct 2020

Dear Dr Chapouly,

On behalf of my colleagues and the Academic Editor, Richard Daneman, I am pleased to inform you that we will be delighted to publish your Research Article in PLOS Biology. 

PRODUCTION PROCESS

Before publication you will see the copyedited word document (within 5 business days) and a PDF proof shortly after that. The copyeditor will be in touch shortly before sending you the copyedited Word document. We will make some revisions at copyediting stage to conform to our general style, and for clarification. When you receive this version you should check and revise it very carefully, including figures, tables, references, and supporting information, because corrections at the next stage (proofs) will be strictly limited to (1) errors in author names or affiliations, (2) errors of scientific fact that would cause misunderstandings to readers, and (3) printer's (introduced) errors. Please return the copyedited file within 2 business days in order to ensure timely delivery of the PDF proof. 

If you are likely to be away when either this document or the proof is sent, please ensure we have contact information of a second person, as we will need you to respond quickly at each point. Given the disruptions resulting from the ongoing COVID-19 pandemic, there may be delays in the production process. We apologise in advance for any inconvenience caused and will do our best to minimize impact as far as possible.

EARLY VERSION

PRESS 

Kind regards,

Erin O'Loughlin

Publishing Editor, 

PLOS Biology

on behalf of

Gabriel Gasque,

Senior Editor

PLOS Biology